# VARIANCE-AWARE SPARSE LINEAR BANDITS

**Yan Dai**
IIIS, Tsinghua University
`yan-dai20@mails.tsinghua.edu.cn`

**Ruosong Wang**
University of Washington
`ruosongw@cs.washington.edu`

**Simon S. Du**
University of Washington
`ssdu@cs.washington.edu`

## ABSTRACT

It is well-known that for sparse linear bandits, when ignoring the dependency on sparsity which is much smaller than the ambient dimension, the worst-case minimax regret is $\widetilde{\Theta}\left(\sqrt{dT}\right)$ where $d$ is the ambient dimension and $T$ is the number of rounds. On the other hand, in the benign setting where there is no noise and the action set is the unit sphere, one can use divide-and-conquer to achieve $\widetilde{\mathcal{O}}(1)$ regret, which is (nearly) independent of $d$ and $T$. In this paper, we present the first variance-aware regret guarantee for sparse linear bandits: $\widetilde{\mathcal{O}}\left(\sqrt{d\sum_{t=1}^{T}\sigma_t^2} + 1\right)$, where $\sigma_t^2$ is the variance of the noise at the $t$-th round. This bound naturally interpolates the regret bounds for the worst-case constant-variance regime (i.e., $\sigma_t \equiv \Omega(1)$) and the benign deterministic regimes (i.e., $\sigma_t \equiv 0$). To achieve this variance-aware regret guarantee, we develop a general framework that converts any variance-aware linear bandit algorithm to a variance-aware algorithm for sparse linear bandits in a "black-box" manner. Specifically, we take two recent algorithms as black boxes to illustrate that the claimed bounds indeed hold, where the first algorithm can handle unknown-variance cases and the second one is more efficient.

## 1 INTRODUCTION

This paper studies the sparse linear stochastic bandit problem, which is a special case of linear stochastic bandits. In linear bandits (Dani et al., 2008), the agent is facing a sequential decision-making problem lasting for $T$ rounds. For the $t$-th round, the agent chooses an action $x_t \in \mathcal{X} \subseteq \mathbb{R}^d$, where $\mathcal{X}$ is an action set, and receives a noisy reward $r_t = \langle \theta^*, x_t \rangle + \eta_t$ where $\theta^* \in \mathcal{X}$ is the (hidden) parameter of the game and $\eta_t$ is random zero-mean noise. The goal of the agent is to minimize her *regret* $\mathcal{R}_T$, that is, the difference between her cumulative reward $\sum_{t=1}^{T} \langle \theta^*, x_t \rangle$ and $\max_{x \in \mathcal{X}} \sum_{t=1}^{T} \langle \theta^*, x \rangle$ (check Eq. (1) for a definition). Dani et al. (2008) proved that the minimax optimal regret for linear bandits is $\widetilde{\Theta}(d\sqrt{T})$ when the noises are independent Gaussian random variables with means 0 and variances 1 and both $\theta^*$ and the actions $x_t$ lie in the unit sphere in $\mathbb{R}^d$.[1]

In real-world applications such as recommendation systems, only a few features may be relevant despite a large candidate feature space. In other words, the high-dimensional linear regime may actually allow a low-dimensional structure. As a result, if we still use the linear bandit model, we will always suffer $\Omega(d\sqrt{T})$ regret no matter how many features are useful. Motivated by this, the sparse linear stochastic bandit problem was introduced (Abbasi-Yadkori et al., 2012; Carpentier & Munos, 2012). This problem has an additional constraint that the hidden parameter, $\theta^*$, is sparse, i.e., $\|\theta^*\|_0 \le s$ for some $s \ll d$. However, the agent has *no* prior knowledge about $s$ and thus the interaction protocol is exactly the same as that of linear bandits. The minimax optimal regret for

---

[1]Throughout the paper, we will use the notations $\widetilde{\mathcal{O}}(\cdot)$ and $\widetilde{\Theta}(\cdot)$ to hide $\log T, \log d, \log s$ (where $s$ is the sparsity parameter, which will be introduced later) and $\log\log\frac{1}{\delta}$ factors (where $\delta$ is the failure probability).

sparse linear bandits is $\widetilde{\Theta}(\sqrt{sdT})$ (Abbasi-Yadkori et al., 2012; Antos & Szepesvári, 2009).[2] This bound bypasses the $\Omega(d\sqrt{T})$ lower bound for linear bandits as we always have $s = \|\theta^*\|_0 \leq d$ and the agent does not have access to $s$ either (though a few previous works assumed a known $s$).

However, both the $\widetilde{\mathcal{O}}(d\sqrt{T})$ and the $\widetilde{\mathcal{O}}(\sqrt{sdT})$ bounds are the *worst-case* regret bounds and sometime are too pessimistic especially when $d$ is large. On the other hand, many problems with delicate structures permit a regret bound much smaller than the worst-case bound. The structure this paper focuses on is the magnitude of the noise. Consider the following motivating example.

**Motivating Example (Deterministic Sparse Linear Bandits).** Consider the case where the action set is the unit sphere $\mathcal{X} = \mathbb{S}^{d-1}$, and there is no noise, i.e., the feedback is $r_t = \langle \theta^*, x_t \rangle$ for each round $t \in [T]$. In this case, one can identify all non-zero entries of $\theta^*$ coordinates in $\mathcal{O}(s \log d)$ steps with high probability via a divide-and-conquer algorithm, and thus yield a *dimension-free* regret $\widetilde{\mathcal{O}}(s)$ (see Appendix C for more details about this).[3] However, this divide-and-conquer algorithm is specific for deterministic sparse linear bandit problems and does not work for noisy models. Henceforth, we study the following natural question:

> *Can we design an algorithm whose regret adapts to the noise level such that the regret interpolates the $\sqrt{dT}$-type bound in the worst case and the dimension-free bound in the deterministic case?*

Before introducing our results, we would like to mention that there are recent works that studied the noise-adaptivity in linear bandits (Zhou et al., 2021; Zhang et al., 2021; Kim et al., 2021). They gave *variance-aware* regret bounds of the form $\widetilde{\mathcal{O}}\left(\mathrm{poly}(d)\sqrt{\sum_{t=1}^{T}\sigma_t^2} + \mathrm{poly}(d)\right)$ where $\sigma_t^2$ is the (conditional) variance of the noise $\eta_t$. This bound reduces to the standard $\widetilde{\mathcal{O}}(\mathrm{poly}(d)\sqrt{T})$ bound in the worst-case when $\sigma_t = \Omega(1)$, and to a constant-type regret $\widetilde{\mathcal{O}}(\mathrm{poly}(d))$ that is independent of $T$. However, compared with the linear bandits setting, the variance-aware bound for sparse linear bandits is more significant because it reduces to a *dimension-free* bound in the noiseless setting. Despite this, to our knowledge, no variance-aware regret bounds exist for sparse linear bandits.

## 1.1 Our Contributions

This paper gives the first set of variance-aware regret bounds for sparse linear bandits. We design a general framework, VASLB, to reduce variance-aware sparse linear bandits to variance-aware linear bandits with little overhead in regret. For ease of presentation, we define the following notation to characterize the *variance-awareness* of a sparse linear bandit algorithm:

**Definition 1.** *A variance-aware sparse linear bandit algorithm $\mathcal{F}$ is $(f(s,d), g(s,d))$-variance-aware, if for any given failure probability $\delta > 0$, with probability $1 - \delta$, $\mathcal{F}$ ensures*

$$\mathcal{R}_T^{\mathcal{F}} \leq \widetilde{\mathcal{O}}\left( f(s,d)\sqrt{\sum_{t=1}^{T}\sigma_t^2\,\mathrm{polylog}\frac{1}{\delta}} + g(s,d)\,\mathrm{polylog}\frac{1}{\delta} \right),$$

*where $\mathcal{R}_T^{\mathcal{F}}$ is the regret of $\mathcal{F}$ in $T$ rounds, $d$ is the ambient dimension and $s$ is the maximum number of non-zero coordinates. Specifically, for linear bandits, $f, g$ are functions only of $d$.*

Hence, an $(f, g)$-variance-aware algorithm will achieve $\widetilde{\mathcal{O}}(f(s,d)\sqrt{T}\,\mathrm{polylog}\frac{1}{\delta})$ worst-case regret and $\widetilde{\mathcal{O}}(g(s,d)\,\mathrm{polylog}\frac{1}{\delta})$ deterministic-case regret. Ideally, we would like $g(s,d)$ being independent of $d$, making the bound *dimension-free* in deterministic cases, as the divide-and-conquer approach.

In this paper, we provide a general framework that can convert *any* linear bandit algorithm $\mathcal{F}$ to a corresponding sparse linear bandit algorithm $\mathcal{G}$ in a black-box manner. Moreover, it is *variance-aware-preserving*, in the sense that, if $\mathcal{F}$ enjoys the variance-aware property, so does $\mathcal{G}$. Generally

---

[2]Carpentier & Munos (2012) and Lattimore et al. (2015) obtained an $\mathcal{O}(s\sqrt{T})$ regret bound under different models. The former one assumed a component-wise noise model, while the latter one assumed a $\|\theta^*\|_1 \leq 1$ ground-truth as well as a $\|x_t\|_\infty \leq 1$ action space. See Appendix A for more discussions on this.

[3]We also remark that some assumptions on the action is needed. For example, if every action can only query one coordinate (each action corresponds to one vector of the standard basis) then an $\Omega(d)$ regret lower bound is unavoidable. Hence, in this paper, we only consider the benign case that action set is the unit sphere.

speaking, if the plug-in linear bandit algorithm $\mathcal{F}$ is $(f(d), g(d))$-variance-aware, then our framework directly gives an $(s(f(s) + \sqrt{d}), s(g(s) + 1))$-variance-aware algorithm $\mathcal{G}$ for sparse linear bandits.

Besides presenting our framework, we also illustrate its usefulness by plugging in two existing variance-aware linear bandit algorithms, where the first one is variance-aware (i.e., works in unknown-variance cases) but computationally inefficient. In contrast, the second one is efficient but requires the variance $\sigma_t^2$ to be delivered together with feedback $r_t$. Their regret guarantees are stated as follows.

1. The first variance-aware linear bandit algorithm we plug in is VOFUL, which was proposed by Zhang et al. (2021) and improved by Kim et al. (2021). This algorithm is computationally inefficient but deals with *unknown variances*. Using this VOFUL, our framework generates a $(s^{2.5} + s\sqrt{d}, s^3)$-variance-aware algorithm for sparse linear bandits. Compared to the $\Omega(\sqrt{sdT})$ regret lower-bound for sparse linear bandits (Lattimore & Szepesvári, 2020, §24.3), our worst-case regret bound is near-optimal up to a factor $\sqrt{s}$. Moreover, our bound is independent of $d$ and $T$ in the deterministic case, nearly matching the bound of divide-and-conquer algorithm dedicated to the deterministic setting up to poly$(s)$ factors.
2. The second algorithm we plug in is Weighted OFUL (Zhou et al., 2021), which requires known variances but is computationally efficient. We obtain an $(s^2 + s\sqrt{d}, s^{1.5}\sqrt{T})$-variance-aware *efficient* algorithm. In the deterministic case, this algorithm can only achieve a $\sqrt{T}$-type regret bound (albeit still independent of $d$). We note that this is not due to our framework but due to Weighted OFUL which itself cannot gives constant regret bound in the deterministic setting.

Moreover, we would like to remark that our deterministic regret can be further improved if a better variance-aware linear bandit algorithm is deployed: The current ones either have $\widetilde{\mathcal{O}}(d^2)$ (Kim et al., 2021) or $\widetilde{\mathcal{O}}(\sqrt{dT})$ (Zhou et al., 2021) regret in the deterministic case, which are both sub-optimal compared with the $\Omega(d)$ lower bound.

## 1.2 RELATED WORK

**Linear Bandits.** This problem was first introduced by Dani et al. (2008), where an algorithm with regret $\mathcal{O}(d\sqrt{T}(\log T)^{3/2})$ and a near-matching regret lower-bound $\Omega(d\sqrt{T})$ were given. After that, an improved upper bound $\mathcal{O}(d\sqrt{T}\log T)$ (Abbasi-Yadkori et al., 2011) together with an improved lower bound $\Omega(d\sqrt{T\log T})$ (Li et al., 2019) were derived. An extension of it, namely linear contextual bandits, where the action set allowed for each step can vary with time (Chu et al., 2011; Kannan et al., 2018; Li et al., 2019; 2021), is receiving more and more attention. The best-arm identification problem where the goal of the agent is to approximate $\theta^*$ with as few samples as possible (Soare et al., 2014; Degenne et al., 2019; Jedra & Proutiere, 2020; Alieva et al., 2021) is also of great interest.

**Sparse Linear Bandits.** Abbasi-Yadkori et al. (2011) and Carpentier & Munos (2012) concurrently considered the sparse linear bandit problem, where the former work assumed a noise model of $r_t = \langle x_t, \theta^* \rangle + \eta_t$ such that $\eta_t$ is $R$-sub-Gaussian and achieved $\widetilde{\mathcal{O}}(R\sqrt{sdT})$ regret, while the latter one considered the noise model of $r_t = \langle x_t + \eta_t, \theta^* \rangle$ such that $\|\eta_t\|_2 \leq \sigma$ and $\|\theta^*\|_2 \leq \theta$, achieving $\widetilde{\mathcal{O}}((\sigma + \theta)^2 s\sqrt{T})$ regret. Lattimore et al. (2015) assumed an hypercube (i.e., $\mathcal{X} = [-1, 1]^d$) action set and a $\|\theta^*\|_1 \leq 1$ ground-truth, yielding $\widetilde{\mathcal{O}}(s\sqrt{T})$ regret. Antos & Szepesvári (2009) proved a $\Omega(\sqrt{dT})$ lower-bound when $s = 1$ with the unit sphere as $\mathcal{X}$. Some recent works considered data-poor regimes where $d \gg T$ (Hao et al., 2020; 2021a;b; Wang et al., 2020), which is beyond the scope of this paper. Another work worth mentioning is the recent work by Dong et al. (2021), which studies bandits or MDPs with deterministic rewards. Their result implies an $\widetilde{\mathcal{O}}(T^{15/16}s^{1/16})$ bound for deterministic sparse linear bandits, which is independent of $d$. They also provided an ad-hoc divide-and-conquer algorithm, which achieves $\mathcal{O}(s \log d)$ regret only for deterministic cases.

**Variance-Aware Online Learning.** For tabular MDPs, the variance information is widely used in both discounted settings (Lattimore & Hutter, 2012) and episodic settings (Azar et al., 2017; Jin et al., 2018), where Zanette & Brunskill (2019) used variance information to derive problem-dependent regret bounds for tabular MDPs. For bandits, Audibert et al. (2009) made use of variance information in multi-armed bandits, giving an algorithm outperforming existing ones when the variances for suboptimal arms are relatively small. For bandits with high-dimensional structures, Faury et al. (2020) studied variance adaptation for logistic bandits, Zhou et al. (2021) considered linear bandits and linear mixture MDPs where the variance information is revealed to the agent,

Table 1: An overview of the proposed algorithms/results and comparisons with related works.

| Algorithm | Setting | Worst-case Regret [a] | Deterministic-case Regret [b] | Efficiency | Variances |
|---|---|---|---|---|---|
| ConfidenceBall$_2$ (Dani et al., 2008) | LinBandit | $\widetilde{\mathcal{O}}(d\sqrt{T})$ | | ✓ | |
| OFUL (Abbasi-Yadkori et al., 2012) | Sparse LinBandit | $\widetilde{\mathcal{O}}(\sqrt{sdT})$ | N/A | ✓ | N/A |
| SL-UCB (Carpentier & Munos, 2012) | Sparse LinBandit | $\widetilde{\mathcal{O}}(s\sqrt{T})$ [c] | | ✓ | |
| Lattimore et al. (2015, Algorithm 4) | Sparse LinBandit | $\widetilde{\mathcal{O}}(s\sqrt{T})$ [d] | | ✓ | |
| Weighted OFUL (Zhou et al., 2021) | LinBandit | $\widetilde{\mathcal{O}}(d\sqrt{T})$ | $\widetilde{\mathcal{O}}(\sqrt{dT})$ | ✓ | Known |
| VOFUL2 (Kim et al., 2021) | LinBandit | $\widetilde{\mathcal{O}}(d^{1.5}\sqrt{T})$ | $\widetilde{\mathcal{O}}(d^2)$ | ✗ | Unknown |
| VASLB (**This work**) | Sparse LinBandit | $\widetilde{\mathcal{O}}(s^2\sqrt{T}+s\sqrt{dT})$ | $\widetilde{\mathcal{O}}(s^{1.5}\sqrt{T})$ | ✓ | Known |
| | Sparse LinBandit | $\widetilde{\mathcal{O}}(s^{2.5}\sqrt{T}+s\sqrt{dT})$ | $\widetilde{\mathcal{O}}(s^3)$ | ✗ | Unknown |
| Lower Bound (Antos & Szepesvári, 2009) | Sparse LinBandit | $\Omega(\sqrt{dT})$ [e] | N/A | N/A | N/A |

[a]"Worst-case" means the variances $\sigma_t^2$ are all 1. Here, $d$ is the ambient dimension, $T$ is the number of rounds, and $s$ is the sparsity parameter (only applicable to sparse linear bandits).

[b]"Deterministic-case" means the variances $\sigma_t^2$ are all 0. Only applicable to variance-aware algorithms.

[c]With a different feedback model; see Appendix A for more comparison.

[d]With a different action set and an different assumption on $\theta^*$; see Appendix A for more comparison.

[e]This bound holds even if $s = 1$ and the action set is fixed to be the unit sphere.

giving an $\widetilde{\mathcal{O}}(d\sqrt{\sum_{t=1}^{T}\sigma_t^2}+\sqrt{dT})$ guarantee for linear bandits, and Zhang et al. (2021) proposed another algorithm for linear bandits and linear mixture MDPs, which does not require any variance information, whose regret can be improved to be $\widetilde{\mathcal{O}}(d^{1.5}\sqrt{\sum_{t=1}^{T}\sigma_t^2}+d^2)$ as shown by Kim et al. (2021). The recent work by Hou et al. (2022) considered variance-constrained best arm identification, where the feedback noise only depends on the action by the agent (whereas ours can depend on time, which is more general than theirs). Another recent work (Zhao et al., 2022) studied variance-aware regret bounds for bandits with general function approximation in the known variance case.

**Stochastic Contextual Sparse Linear Bandits.** In the setting, the action set for each round $t$ is i.i.d. sampled (called the "context"). It is known that $\widetilde{\mathcal{O}}(\sqrt{sT})$ regret is achievable in this setting (Kim & Paik, 2019; Ren & Zhou, 2020; Oh et al., 2021; Ariu et al., 2022). However, in our setting where both action set $\mathcal{X}$ and ground-truth $\theta^*$ are fixed, a polynomial dependency on $d$ is in general unavoidable because it is impossible to learn more than one parameter per arm (Bastani & Bayati, 2020), agreeing with the $\Omega(\sqrt{dT})$ lower bound when $s = 1$ (Antos & Szepesvári, 2009; Abbasi-Yadkori et al., 2012).

## 2 PROBLEM SETUP

**Notations.** We use $[N]$ to denote the set $\{1, 2, \ldots, N\}$ where $N \in \mathbb{N}$. For a vector $x \in \mathbb{R}^d$, we use $\|x\|_p$ to its $L_p$-norm, namely $\|x\|_p \triangleq (\sum_{i=1}^{d} x_i^p)^{1/p}$. We use $\mathbb{S}^{d-1}$ to denote the $(d-1)$-dimensional unit sphere, i.e., $\mathbb{S}^{d-1} \triangleq \{x \in \mathbb{R}^d \mid \|x\|_2 = 1\}$. We use $\widetilde{\mathcal{O}}(\cdot)$ and $\widetilde{\Theta}(\cdot)$ to hide all logarithmic factors in $T, s, d$ and $\log\frac{1}{\delta}$ (see Footnote 1). For a random event $\mathcal{E}$, we denote its indicator by $\mathbb{1}[\mathcal{E}]$.

We assume the action space and the ground-truth space are both the $(d-1)$-dimensional unit sphere, denoted by $\mathcal{X} \triangleq \mathbb{S}^{d-1}$. Denote the ground-truth by $\theta^* \in \mathcal{X}$. There will be $T \geq 1$ rounds for the agent to make decisions sequentially. At the beginning of round $t \in [T]$, the agent has to choose an action $x_t \in \mathcal{X}$. At the end of step $t$, the agent receives a *noisy* feedback $r_t = \langle x_t, \theta^* \rangle + \eta_t, \forall t \in [T]$, where $\eta_t$ is an independent zero-mean Gaussian random variable. Denote by $\sigma_t^2 = \text{Var}(\eta_t)$ the variance of $\eta_t$. For a fair comparison with non-variance-aware algorithms, we assume that $\sigma_t^2 \leq 1$. The agent then receives a (deterministic and unrevealed) reward of magnitude $\langle x_t, \theta^* \rangle$ for this round.

The agent is allowed to make the decision $x_t$ based on all historical actions $x_1, \ldots, x_{t-1}$, all historical feedback $r_1, \ldots, r_{t-1}$, and any amount of private randomness. The agent's goal is to minimize the regret, defined as follows.

**Definition 2** (Regret). *The following random variable is the **regret** of a linear bandit algorithm:*

$$\mathcal{R}_T = \max_{x \in \mathcal{X}} \sum_{t=1}^{T} \langle x, \theta^* \rangle - \sum_{t=1}^{T} \langle x_t, \theta^* \rangle = \sum_{t=1}^{T} \langle \theta^* - x_t, \theta^* \rangle, \tag{1}$$

*where the second equality is due to our assumption that $\mathcal{X} = \mathbb{S}^{d-1}$.*

---

**Algorithm 1** Variance-Aware Sparse Linear Bandits (`VASLB`) Framework

---

**Input:** Number of dimensions $d$, linear bandit algorithm $\mathcal{F}$ and its regret estimator $\overline{\mathcal{R}_n^{\mathcal{F}}}$

1: Initialize gap threshold $\Delta \leftarrow \frac{1}{4}$, estimated "good" coordinates $S \leftarrow \varnothing$, current round $t \leftarrow 0$.

2: **while** $t < T$ **do** $\qquad\qquad\qquad\qquad$ ▷ The algorithm automatically increases $t$ by 1 per query.

3: $\quad$ **if** $S \neq \varnothing$ **then** $\qquad\qquad\qquad\qquad\qquad\qquad$ ▷ Have some coordinates to "commit".

4: $\qquad$ Initialize a new linear bandit instance $\mathcal{F}$ on coordinates $S$. $\qquad$ ▷ "Commit" phase.

5: $\qquad$ Execute $\mathcal{F}$ for $n_\Delta^a \geq 1$ steps & maintain pessimistic estimation $\overline{\mathcal{R}_{n_\Delta^a}^{\mathcal{F}}}$, until $\frac{1}{n_\Delta^a}\overline{\mathcal{R}_{n_\Delta^a}^{\mathcal{F}}} < \Delta^2$.

6: $\qquad$ Suppose that $\mathcal{F}$ plays $x_1, x_2, \ldots, x_{n_\Delta^a}$. Set $\widehat{\theta} = \frac{1}{n_\Delta^a}\sum_{i=1}^{n_\Delta^a} x_i$ as the estimate for $\{\theta_i^*\}_{i \in S}$.

7: $\quad$ **if** $\sum_{i \in S} \widehat{\theta}_i^2 \leq 1 - \Delta^2$ **then** $\qquad\qquad$ ▷ Still have undiscovered coordinates with $\theta_i^* > \frac{\Delta}{2}$

8: $\qquad$ Let $R \leftarrow \sqrt{1 - \sum_{i \in S} \widehat{\theta}_i^2}$, $K = d - |S|$. $\qquad\qquad\qquad\qquad$ ▷ "Explore" phase.

9: $\qquad$ Perform $n_\Delta^b \geq 1$ calls to RANDOMPROJECTION$(K, R, S, \widehat{\theta})$ in Algorithm 2, until

$$2\sqrt{2\sum_{k=1}^{n_\Delta^b}(r_{k,i} - \overline{r}_i)^2 \ln\frac{4}{\delta}} < n_\Delta^b \cdot \frac{\Delta}{4}, \quad \forall 1 \leq i \leq K, \tag{2}$$

$\qquad$ where $r_k$ is the $k$-th return vector of RANDOMPROJECTION and $\overline{r} \triangleq \frac{1}{n_\Delta^b}\sum_{k=1}^{n_\Delta^b} r_k$.

10: $\qquad$ **for** $i = 1, 2, \ldots, K$ **do**

11: $\qquad\qquad$ **if** $|\overline{r}_i| > \Delta$ where $\overline{r} = \frac{1}{n_\Delta^b}\sum_{k=1}^{n_\Delta^b} r_k$ **then** add the $i$-th element that is not in $S$ to $S$.

---

**Algorithm 2** The RANDOMPROJECTION Subroutine

---

1: **function** RANDOMPROJECTION$(K, R, S, \widehat{\theta})$

2: $\quad$ Generate $K$ i.i.d. samples $y_1, y_2, \ldots, y_K$, each with equal probability being $\pm\frac{R}{\sqrt{K}}$.

3: $\quad$ Play $x \in \mathcal{X}$ constructed as $x_i = \begin{cases} \widehat{\theta}_i, & i \in S \\ y_j, & i \text{ is the } j\text{-th element that is not in } S \end{cases}$

4: $\quad$ **return** $\frac{K}{R^2}((r - \sum_{i \in S}\widehat{\theta}_i^2)\, y)$ where $r = \langle x, \theta^* \rangle + \eta$ is the (noisy) feedback.

---

For the sparse linear bandit problem, we have an additional restriction that $\|\theta^*\|_0 \leq s$, i.e., there are at most $s$ coordinates of $\theta^*$ is non-zero. However, as mentioned in the introduction, the agent does *not* know anything about $s$ – she only knows that she is facing a (probably sparse) linear environment.

## 3 FRAMEWORK AND ANALYSIS

Our framework `VASLB` is presented in Algorithm 1. We explain its design in Section 3.1 and sketch its analysis in Section 3.2. Then we give two applications using `VOFUL2` (Kim et al., 2021) and `Weighted VOFUL` (Zhou et al., 2021) as $\mathcal{F}$, whose analyses are sketched in Sections 4.1 and 4.2.

### 3.1 MAIN DIFFICULTIES AND TECHNICAL OVERVIEW

At a high level, our framework follows the spirit of the classic "explore-then-commit" approach (which is directly adopted by Carpentier & Munos (2012)), where the agent first identifies those "huge" entries of $\theta^*$ and then performs a linear bandit algorithm on them. However, it is hard to incorporate variances into this vanilla idea to make it variance-aware – the desired regret depends on variances and is thus unknown to the agent. Thus it is difficult to determine a "gap threshold" $\Delta$ (that is, the agent stops to "commit" after identifying all $\theta_i^* \geq \Delta$) within a few rounds. For example, in the deterministic case, the agent must identify all non-zero entries to make the regret independent of $T$; on the other hand, in the worst case where $\sigma_t \equiv 1$, the agent only needs to identify all entries with magnitude at least $T^{-1/4}$ to yield $\sqrt{T}$-style regret bounds. At the same time, the actual setting might be mixture of them (e.g., $\sigma_t \equiv 0$ for $t \leq t_0$ and $\sigma_t \equiv 1$ for $t > t_0$ where $t_0 \in [T]$). As a result, such an idea cannot always succeed in determining the correct threshold $\Delta$ and getting the desired regret.

In our proposed framework, we tackle this issue by "explore-then-commit" multiple times. We reduce the uncertainty gently and alternate between "explore" and "commit" modes. We decrease a "gap threshold" $\Delta$ in a halving manner and, at the same time, maintain a set $S$ of coordinates that we believe to have a magnitude larger than $\Delta$. For each $\Delta$, we "explore" (estimating $\theta_i^*$ and adding those greater than $\Delta$ into $S$) and "commit" (performing linear bandit algorithms on coordinates in $S$).

However, as we "explore" again after "committing", we face a unique challenge: Suppose that some entry $i \in [d]$ is identified to be at least $2\Delta$ by previous "explore" phases. During the next "explore" phase, we cannot directly do pure exploration over the remaining unidentified coordinates – otherwise, coordinate $i$ will incur $4\Delta^2$ regret for each round. Fortunately, we can get an estimation $\widehat{\theta}_i$ of $\theta_i^*$ during the previous "commit" phase thanks to the regret-to-sample-complexity conversion (Eq. (3)). Guarded with this estimation, we can reserve $\widehat{\theta}_i$ mass for arm $i$ and subtract $\widehat{\theta}_i^2$ from the feedback in subsequent "explore" phases. More preciously, we do the following.

1. In the "commit" phase where we apply the black-box $\mathcal{F}$, we estimate $\{\theta_i^*\}_{i \in S}$ by the regret-to-sample-complexity conversion: Suppose $\mathcal{F}$ plays $x_1, x_2, \ldots, x_n$ and achieves regret $\mathcal{R}_n^{\mathcal{F}}$, then

$$\langle \theta^* - \widehat{\theta}, \theta^* \rangle \leq \frac{\mathcal{R}_n^{\mathcal{F}}}{n}, \text{ where } \widehat{\theta} \triangleq \frac{1}{n} \sum_{i=1}^n x_i. \tag{3}$$

   Hence, if we take $\{\widehat{\theta}_i\}_{i \in S}$ as an estimate of $\{\theta_i^*\}_{i \in S}$, the estimation error shrinks as $\mathcal{R}_n^{\mathcal{F}}$ is sublinear and the LHS of Eq. (3) is non-negative. Moreover, as we can show that $\widehat{\theta}$ is not away from $\mathcal{X}$ by a lot (Lemma 18), we can safely use $\{\widehat{\theta}_i\}_{i \in S}$ to estimate $\{\theta_i^*\}_{i \in S}$ in subsequent phases. More importantly, if we are granted access to $\mathcal{R}_n^{\mathcal{F}}$, we know how close the estimate is; we can proceed to the next stage once it becomes satisfactory. But it is unrevealed. Fortunately, we know the regret guarantee of $\mathcal{F}$, namely $\overline{\mathcal{R}_n^{\mathcal{F}}}$, which can serve as a *pessimistic* estimation of $\mathcal{R}_n^{\mathcal{F}}$. Hence, terminating when $\frac{1}{n}\overline{\mathcal{R}_n^{\mathcal{F}}} < \Delta^2$ can ensure $\langle \theta^* - \widehat{\theta}, \theta^* \rangle < \Delta^2$ to hold with high probability.
2. In the "exploration" phase, as mentioned before, we can keep the regret incurred by the coordinates identified in $S$ small by putting mass $\widehat{\theta}_i$ for each $i \in S$. For the remaining ones, we use random projection, an idea borrowed from compressed sensing literature (Blumensath & Davies, 2009; Carpentier & Munos, 2012), to find those with large magnitudes to add them to $S$.

   One may notice that putting mass $\widehat{\theta}_i$ for all $i \in S$ will induce bias to our estimation as $\sum_{i \in S} \widehat{\theta}_i^2 \neq \sum_{i \in S} \widehat{\theta}_i \theta_i^*$. However, as $\widehat{\theta}_i$ is close to $\theta_i^*$, this bias will be bounded by $\mathcal{O}(\Delta^2)$ and become dominated by $\frac{\Delta}{4}$ as $\Delta$ decreases. Hence, if we omit this bias, we can overestimate the estimation error due to standard concentration inequalities like Empirical Bernstein (Maurer & Pontil, 2009; Zhang et al., 2021). Once it becomes small enough, we alternate to the "commit" phase again.

Therefore, with high probability, we can ensure all coordinates not in $S$ have magnitudes no more than $\mathcal{O}(\Delta)$ and all coordinates in $S$ will together contribute regret bounded by $\mathcal{O}(\Delta^2)$. Hence, the regret in each step is (roughly) bounded by $\mathcal{O}(s\Delta^2)$. Upper bounding the number of steps needed for each stage and exploiting the regret guarantees of the chosen $\mathcal{F}$ then gives well-bounded regret.

## 3.2 ANALYSIS OF THE FRAMEWORK

**Notations.** For each $\Delta$, let $\mathcal{T}_{\Delta}$ be the set of rounds associated with $\Delta$. By our algorithm, each $\mathcal{T}_{\Delta}$ should be an interval. Moreover, $\{\mathcal{T}_{\Delta}\}_{\Delta}$ forms a partition of $[T]$. Define $\mathcal{T}_{\Delta}^a$ as all the rounds in the "commit" phase when the gap threshold is $\Delta$ (where $\mathcal{F}$ is executed), and $\mathcal{T}_{\Delta}^b$ as the "explore" phase (i.e., those executing RANDOMPROJECTION). Let $\widetilde{\mathcal{T}_{\Delta}^a}$ and $\widetilde{\mathcal{T}_{\Delta}^b}$ be the steps that the agent decided not to proceed in $\mathcal{T}_{\Delta}^a$ and $\mathcal{T}_{\Delta}^b$, respectively, which are formally defined as $\widetilde{\mathcal{T}_{\Delta}^i} = \{t \in \mathcal{T}_{\Delta}^i \mid t \neq \max_{t' \in \mathcal{T}_{\Delta}^i} t'\}$, $i = a, b$. Define the final value of $\Delta$ as $\Delta_f$. Denote $n_{\Delta}^a = |\mathcal{T}_{\Delta}^a|$ and $n_{\Delta}^b = |\mathcal{T}_{\Delta}^b|$ (both are stopping times). We have $\sum_{\Delta = 2^{-2}, \ldots, \Delta_f} (n_{\Delta}^a + n_{\Delta}^b) = T$. We can then decompose $\mathcal{R}_T$ into $\mathcal{R}_T^a$ and $\mathcal{R}_T^b$:

$$\mathcal{R}_T^a = \sum_{\Delta = 2^{-2}, \ldots, \Delta_f} \sum_{t \in \mathcal{T}_{\Delta}^a} \langle \theta^* - x_t, \theta^* \rangle, \quad \mathcal{R}_T^b = \sum_{\Delta = 2^{-2}, \ldots, \Delta_f} \sum_{t \in \mathcal{T}_{\Delta}^b} \langle \theta^* - x_t, \theta^* \rangle,$$

where $\mathcal{R}_T^a$ may depend on the choice of $\mathcal{F}$ and $\mathcal{R}_T^b$ only depends on the framework (Algorithm 1) itself. We now show that, as long as the regret estimation $\overline{\mathcal{R}_n^{\mathcal{F}}}$ is indeed an overestimation of $\mathcal{R}_n^{\mathcal{F}}$

with high probability, we can get a good upper bound of $\mathcal{R}_T^b$, which is formally stated as Theorem 3. The full proof of Theorem 3 will be presented in Appendix F and is only sketched here.

**Theorem 3.** *Suppose that for any execution of $\mathcal{F}$ that last for $n$ steps, $\overline{\mathcal{R}_n^{\mathcal{F}}} \geq \mathcal{R}_n^{\mathcal{F}}$ holds with probability $1 - \delta$, i.e., $\overline{\mathcal{R}_n^{\mathcal{F}}}$ is pessimistic. Then the total regret incurred by the second phase satisfies*

$$\mathcal{R}_T^b = \widetilde{\mathcal{O}}\left(s\sqrt{d}\sqrt{\sum_{t=1}^{T} \sigma_t^2 \log\frac{1}{\delta}} + s\log\frac{1}{\delta}\right) \quad \text{with probability } 1 - \delta.$$

**Remark.** This theorem indicates that our framework *itself* will only induce an $(s\sqrt{d}, s)$-variance-awareness to the resulting algorithm. As noticed by Abbasi-Yadkori et al. (2011), when $\sigma_t \equiv 1$, $\Omega(\sqrt{sdT})$ regret is unavoidable, which means that it is only sub-optimal by a factor no more than $\sqrt{s}$. Moreover, for deterministic cases, the $\widetilde{\mathcal{O}}(s)$ regret also matches the aforementioned divide-and-conquer algorithm, which is specially designed and can only work for deterministic cases.

*Proof Sketch of Theorem 3.* We define two good events with high probability for a given gap threshold $\Delta$: $\mathcal{G}_\Delta$ and $\mathcal{H}_\Delta$. Informally, $\mathcal{G}_\Delta$ means $\sum_{i \in S} \theta_i^*(\theta_i^* - \widehat{\theta}_i) < \Delta^2$ (i.e., $\widehat{\theta}$ is close to $\theta^*$ after "commit") and $\mathcal{H}_\Delta$ stands for $|\theta_i^*| \geq \Omega(\Delta)$ if and only if $i \in S$ (i.e., we "explore" correctly). Check Eq. (10) in the appendix for formal definitions. For $\mathcal{G}_\Delta$, from Eq. (3), we know that it happens as long as $\overline{\mathcal{R}_n^{\mathcal{F}}} \geq \mathcal{R}_n^{\mathcal{F}}$. It remains to argue that $\Pr\{\mathcal{H}_\Delta \mid \mathcal{G}_\Delta, \mathcal{H}_{2\Delta}\} \geq 1 - s\delta$.

By Algorithm 2, the $i$-th coordinate of each $r_k$ ($1 \leq k \leq n_\Delta^b$) is an independent sample of

$$\frac{K}{R^2}(y_i)^2 \theta_i^* + \sum_{j \in S} \widehat{\theta}_j(\theta_j^* - \widehat{\theta}_j) + \sum_{j \notin S, j \neq i}\left(\frac{K}{R^2}y_i y_j\right)\theta_j^* + \left(\frac{K}{R^2}y_i\right)\eta_n, \tag{4}$$

where $\frac{\sqrt{K}}{R}y_i$ is an independent Rademacher random variable. After conditioning on $\mathcal{G}_\Delta$ and $\mathcal{H}_{2\Delta}$, $\sum_{i \in S} \widehat{\theta}_i^2$ and $\sum_{i \in S} \widehat{\theta}_i \theta_i^*$ will be close. Therefore, the first term is exactly $\theta_i^*$ (the magnitude we want to estimate), the second term is a small bias bounded by $\mathcal{O}(\Delta^2)$ and the last two terms are zero-mean noises, which are bounded by $\frac{\Delta}{4}$ according to Empirical Bernstein Inequality (Theorem 10) and our choice of $n_\Delta^b$ (Eq. (2)). Hence, $\Pr\{\mathcal{H}_\Delta \mid \mathcal{G}_\Delta, \mathcal{H}_{2\Delta}\} \geq 1 - s\delta$.

Let us focus on an arm $i^*$ never identified into $S$ in Algorithm 1. By definition of $n_\Delta^b$ (Eq. (2)),

$$(n_\Delta^b - 1)\frac{\Delta}{4} < 2\sqrt{2\sum_{t \in \widetilde{\mathcal{T}}_\Delta^b}(r_{t,i^*} - \overline{r}_{i^*})^2 \ln\frac{4}{\delta}} \leq 2\sqrt{2\sum_{t \in \widetilde{\mathcal{T}}_\Delta^b}(r_{t,i^*} - \mathbb{E}[r_{t,i^*}])^2 \ln\frac{4}{\delta}},$$

where the second inequality is due to properties of sample variances. By $\mathcal{G}_\Delta$, those coordinates in $S$ will incur regret of $\sum_{i \in S}(\theta_i^* - x_{t,i})\theta_i^* = \sum_{i \in S}(\theta_i^* - \widehat{\theta}_i)\theta_i^* < \Delta^2$ for all $t \in \mathcal{T}_\Delta^b$. Moreover, by $\mathcal{H}_{2\Delta}$, each arm outside $S$ will roughly incur $n_\Delta^b(\theta_i^*)^2 = \mathcal{O}(n_\Delta^b\Delta^2)$ regret, as $y_i$'s are independent and zero-mean. As there are at most $s$ non-zero coordinates, the total regret for $\mathcal{T}_\Delta^b$ will be roughly bounded by $\mathcal{O}(n_\Delta^b \cdot s\Delta^2)$ (there exists another term due to randomized $y_i$'s, which is dominated and omitted here; check Lemma 21 for more details). Hence, the total regret is bounded by

$$\mathcal{R}_T^b \lesssim \sum_\Delta \mathcal{O}(sn_\Delta^b\Delta^2) = s \cdot \widetilde{\mathcal{O}}\left(\sum_\Delta \Delta\sqrt{\sum_{t \in \mathcal{T}_\Delta^b}(r_{t,i^*} - \mathbb{E}[r_{t,i^*}])^2 \ln\frac{4}{\delta}}\right) + \mathcal{O}(s).$$

To avoid undesired $\text{poly}(T)$ factors, we cannot directly apply Cauchy-Schwartz inequality to the sum of square roots (as there are a lot of $\Delta$'s). Instead, again by definition of $n_\Delta^b$ (Eq. (2)), we observe the following lower bound of $n_\Delta^b$, which holds for all $\Delta$'s except for $\Delta_f$: $n_\Delta^b \geq \mathcal{O}\left(\frac{1}{\Delta}\sqrt{\sum_{t \in \mathcal{T}_\Delta^b}(r_{t,i^*} - \mathbb{E}[r_{t,i^*}])^2 \ln\frac{1}{\delta}}\right)$. As $\sum_\Delta n_\Delta^b \leq T$, some arithmetic calculation gives

(intuitively, by thresholding, we manage to "move" the summation over $\Delta$ into the square root, though suffering an extra logarithmic factor; see Eq. (15) in the appendix for more details)

$$\sum_{\Delta \neq \Delta_f} \Delta \sqrt{\sum_{t \in \mathcal{T}_\Delta^b} (r_{t,i^*} - \mathbb{E}[r_{t,i^*}])^2} = \widetilde{\mathcal{O}} \left( \sqrt{\sum_{\Delta \neq \Delta_f} \Delta^2 \sum_{t \in \mathcal{T}_\Delta^b} (r_{t,i^*} - \mathbb{E}[r_{t,i^*}])^2} \right).$$

For a given $\Delta$ and any $1 \leq k \leq n_\Delta^b$, the expectation of $(r_{k,i^*} - \mathbb{E}[r_{k,i^*}])^2$ is bounded by $\left(1 + \frac{K}{R^2}\sigma_k^2\right)$ (Eq. (19) in the appendix), which is no more than $\left(1 + \frac{4d}{\Delta^2}\sigma_k^2\right)$. By concentration properties in the sample variances (Theorem 14 in the appendix), the empirical $(r_{k,i^*} - \mathbb{E}[r_{k,i^*}])^2$ should also be close to $\left(1 + \frac{4}{d^2}\sigma_k^2\right)$; hence, one can write (omitting all $\log\frac{1}{\delta}$ terms)

$$\mathcal{R}_T^b = \mathcal{O}\left(\sum_\Delta s n_\Delta^b \Delta^2\right) = \widetilde{\mathcal{O}}\left(s\sqrt{\sum_\Delta n_\Delta^b \Delta^2} + s\sqrt{d\sum_{t=1}^T \sigma_t^2}\right).$$

As $\sum_\Delta n_\Delta^b \Delta^2$ appears on both sides, we can apply the "self-bounding" property (Efroni et al., 2020, Lemma 38) to conclude $\mathcal{R}_T^b = \mathcal{O}(\sum_\Delta s n_\Delta^b \Delta^2) = \widetilde{\mathcal{O}}\left(s\sqrt{d\sum_{t=1}^T \sigma_t^2} + s\right)$, as claimed. □

## 4 APPLICATIONS OF THE PROPOSED FRAMEWORK

After showing Theorem 16, it only remains to bound $\mathcal{R}_T^a$, which depends on the choice of the plug-in algorithm $\mathcal{F}$. In this section, we give two specific choices of $\mathcal{F}$, VOFUL2 (Kim et al., 2021) and Weighted OFUL (Zhou et al., 2021). The former algorithm does not require the information of $\sigma_t$'s (i.e., it works in unknown-variance cases), albeit computationally inefficient. In contrast, the latter is computationally efficient but requires $\sigma_t^2$ to be revealed with the feedback $r_t$ at round $t$.

### 4.1 COMPUTATIONALLY INEFFICIENT ALGORITHM FOR UNKNOWN VARIANCES

We first use the VOFUL2 algorithm from Kim et al. (2021) as the plug-in algorithm $\mathcal{F}$, which has the following regret guarantee. Note that this is slightly stronger than the original bound: We derive a strengthened "self-bounding" version of it (the first inequality), which is critical to our analysis.

**Proposition 4** (Kim et al. (2021, Variant of Theorem 2))**.** *VOFUL2 executed for $n$ rounds on $d$ dimensions guarantees, w.p. at least $1 - \delta$, there exists a constant $C = \widetilde{\mathcal{O}}(1)$ such that $\mathcal{R}_n^{\mathcal{F}} \leq C\left(d^{1.5}\sqrt{\sum_{k=1}^n \eta_k^2 \ln\frac{1}{\delta}} + d^2 \ln\frac{4}{\delta}\right) = \widetilde{\mathcal{O}}\left(d^{1.5}\sqrt{\sum_{k=1}^n \sigma_k^2 \log\frac{1}{\delta}} + d^2 \log\frac{4}{\delta}\right)$, where $n$ is a stopping time finite a.s. and $\sigma_1^2, \sigma_2^2, \ldots, \sigma_n^2$ are the variances of the independent Gaussians $\eta_1, \eta_2, \ldots, \eta_n$.*

We now construct the regret over-estimation $\overline{\mathcal{R}_n^{\mathcal{F}}}$. Due to unknown variances, it is not straightforward. Our rescue is to use ridge linear regression $\widehat{\beta} \triangleq \operatorname{argmin}_{\beta \in \mathbb{R}^d} \left(\sum_{k=1}^n (r_k - \langle x_k, \beta \rangle)^2 + \lambda\|\beta\|_2\right)$ for samples $\{(x_k, r_k)\}_{k=1}^n$, which ensures that the empirical variance estimation $\sum_{k=1}^n (r_k - \langle x_k, \widehat{\beta} \rangle)^2$ differs from the true sample variance $\sum_{k=1}^n \eta_k^2 = \sum_{k=1}^n (r_k - \langle x_k, \beta^* \rangle)^2$ by no more than $\widetilde{\mathcal{O}}(s\log\frac{1}{\delta})$ (check Appendix E for a formal version). Accordingly, from Proposition 4, we can see that

$$\mathcal{R}_n^{\mathcal{F}} \leq \overline{\mathcal{R}_n^{\mathcal{F}}} \triangleq C\left(s^{1.5}\sqrt{\sum_{k=1}^n (r_k - \langle x_k, \widehat{\beta} \rangle)^2 \ln\frac{1}{\delta}} + s^2\sqrt{2\ln\frac{n}{s\delta^2}\ln\frac{1}{\delta}} + s^{1.5}\sqrt{2\ln\frac{1}{\delta}} + s^2\ln\frac{1}{\delta}\right). \quad (5)$$

Moreover, one can observe that the total sample variance $\sum_{k=1}^n \eta_k^2$ is bounded by (a constant multiple of) the total variance $\sum_{k=1}^n \sigma_k^2$ (which is formally stated as Theorem 13 in the appendix). Therefore, with Eq. (5) as our pessimistic regret estimation $\overline{\mathcal{R}_n^{\mathcal{F}}}$, we have the following regret guarantee.

**Theorem 5** (Regret of Algorithm 1 with VOFUL2)**.** *Algorithm 1 with VOFUL2 as $\mathcal{F}$ and $\overline{\mathcal{R}_n^{\mathcal{F}}}$ defined in Eq. (5) ensures that $\mathcal{R}_T = \widetilde{\mathcal{O}}\left((s^{2.5} + s\sqrt{d})\sqrt{\sum_{t=1}^T \sigma_t^2 \log\frac{1}{\delta}} + s^3 \log\frac{1}{\delta}\right)$ with probability $1 - \delta$.*

Due to space limitations, we defer the full proof to Appendix G.1 and only sketch it here.

*Proof Sketch of Theorem 5.* To bound $\mathcal{R}_T^a$, we consider the regret from the coordinates in and outside $S$ separately. For the former, the total regret in a single phase with gap threshold $\Delta$ is simply controlled by $\widetilde{\mathcal{O}}\left(s^{1.5}\sqrt{\sum_{t\in\mathcal{T}_\Delta^a}\eta_t^2\log\frac{1}{\delta}}+s^2\log\frac{1}{\delta}\right)$ (thanks to Proposition 4). For the latter, each non-zero coordinate outside $S$ can at most incur $\mathcal{O}(\Delta^2)$ regret for each $t\in\mathcal{T}_\Delta^a$. By definition of $n_\Delta^a$ (Line 5), we have $n_\Delta^a=|\mathcal{T}_\Delta^a|=\mathcal{O}\left(\frac{s^{1.5}}{\Delta^2}\sqrt{\sum_{t\in\widetilde{\mathcal{T}}_\Delta^a}\eta_t^2\ln\frac{1}{\delta}}+\frac{s^2}{\Delta^2}\ln\frac{1}{\delta}\right)$, just like the proof of Theorem 3. As the regret from the second part is bounded by $\mathcal{O}(s\Delta^2\cdot n_\Delta^a)$, these two parts together sum to

$$\mathcal{R}_T^a\le\sum_\Delta\mathcal{O}\left(s^{2.5}\sqrt{\sum_{t\in\mathcal{T}_\Delta^a}\eta_t^2\log\frac{1}{\delta}}+s^3\log\frac{1}{\delta}+s\Delta^2\right).$$

As in Theorem 3, we notice that $n_\Delta^a=\Omega\left(\frac{s^{1.5}}{\Delta^2}\sqrt{\sum_{t\in\widetilde{\mathcal{T}}_\Delta^a}\eta_t^2\ln\frac{1}{\delta}}+\frac{s^2}{\Delta^2}\ln\frac{1}{\delta}\right)$ for all $\Delta\ne\Delta_f$ again by definition of $n_\Delta^a$. This will move the summation over $\Delta$ into the square root. Moreover, by the fact that $\eta_t^2=\mathcal{O}(\sigma_t^2\log\frac{1}{\delta})$ (Theorem 14 in the appendix), we have $\mathcal{R}_T^a=\widetilde{\mathcal{O}}\left(s^{2.5}\sqrt{\sum_{t=1}^T\sigma_t^2}\log\frac{1}{\delta}+s^3\log\frac{1}{\delta}\right)$. Combining this with the bound of $\mathcal{R}_T^b$ provided by Theorem 3 concludes the proof. $\square$

## 4.2 COMPUTATIONALLY EFFICIENT ALGORITHM FOR KNOWN VARIANCES

In this section, we consider a computational efficient algorithm `Weighted OFUL` (Zhou et al., 2021), which itself requires $\sigma_t^2$ to be presented at the end of round $t$. Their algorithm guarantees:

**Proposition 6** (Zhou et al. (2021, Corollary 4.3)). *With probability at least $1-\delta$, `Weighted OFUL` executed for $n$ steps on $d$ dimensions guarantees $\mathcal{R}_T^{\mathcal{F}}\le C(\sqrt{dn\log\frac{1}{\delta}}+d\sqrt{\sum_{k=1}^n\sigma_k^2\log\frac{1}{\delta}})$, where $C=\widetilde{\mathcal{O}}(1)$, $n$ is a stopping time finite a.s., and $\sigma_1^2,\sigma_2^2,\dots,\sigma_n^2$ are the variances of $\eta_1,\eta_2,\dots,\eta_n$.*

Taking $\mathcal{F}$ as `Weighted OFUL`, we will have the following regret guarantee for sparse linear bandits:

**Theorem 7** (Regret of Algorithm 1 with `Weighted OFUL`). *Algorithm 1 with `Weighted OFUL` as $\mathcal{F}$ and $\overline{\mathcal{R}_n^{\mathcal{F}}}$ defined as $C(\sqrt{sn\ln\frac{1}{\delta}}+s\sqrt{\sum_{k=1}^n\sigma_k^2\ln\frac{1}{\delta}})$ guarantees $\mathcal{R}_T=\widetilde{\mathcal{O}}\left((s^2+s\sqrt{d})\sqrt{\sum_{t=1}^T\sigma_t^2}\log\frac{1}{\delta}+s^{1.5}\sqrt{T}\log\frac{1}{\delta}\right)$ with probability $1-\delta$.*

The proof is similar to that of Theorem 5, i.e., bounding $n_\Delta^a$ by Line 5 of Algorithm 1 and then using summation techniques to move the summation over $\Delta$ into the square root. The only difference is that we will need to bound $\mathcal{O}(\sum_\Delta\Delta^{-2})$, which seems to be as large as $T$ if we follow the analysis of Theorem 5. However, as we included an additive factor $\sqrt{sn\ln\frac{1}{\delta}}$ in the regret over-estimation $\overline{\mathcal{R}_n^{\mathcal{F}}}$, we have $n_\Delta^a\ge\Delta^{-2}\sqrt{sn_\Delta^a\ln\frac{1}{\delta}}$, which means $n_\Delta^a=\Omega(s\Delta^{-4})$. From $\sum_\Delta n_\Delta^a\le T$, we can consequently bound $\sum_\Delta\Delta^{-2}$ as $\mathcal{O}(\sqrt{\frac{T}{s}})$. The remaining part is just an analog of Theorem 5. Therefore, the proof is omitted in the main text and postponed to Appendix H.

## 5 CONCLUSION

We considered the sparse linear bandit problem with heteroscedastic noises and provided a general framework to reduce *any* variance-aware linear bandit algorithm $\mathcal{F}$ to an algorithm $\mathcal{G}$ for sparse linear bandits that is also variance-aware. We first applied the computationally inefficient algorithm `VOFUL` from Zhang et al. (2021) and Kim et al. (2021). The resulting algorithm works for the unknown-variance case and gets $\widetilde{\mathcal{O}}((s^{2.5}+s\sqrt{d})\sqrt{\sum_{t=1}^T\sigma_t^2}\log\frac{1}{\delta}+s^3\log\frac{1}{\delta})$ regret, which, when regarding the sparsity factor $s\ll d$ as a constant, not only is worst-case optimal but also enjoys constant regret in deterministic cases. We also applied the efficient algorithm `Weighted OFUL` by Zhou et al. (2021) that requires known variance; we got $\widetilde{\mathcal{O}}((s^2+s\sqrt{d})\sqrt{\sum_{t=1}^T\sigma_t^2}\log\frac{1}{\delta}+(\sqrt{sT}+s)\log\frac{1}{\delta})$ regret, still independent of $d$ in deterministic cases. See Appendix B for several future directions.

ACKNOWLEDGMENTS

We greatly acknowledge Csaba Szepesvári for sharing the manuscript (Antos & Szepesvári, 2009) with us, which shows the $\Omega(\sqrt{dT})$ lower bound for sparse linear bandits when $s = 1$ and the action sets are unit balls. The authors thank Xiequan Fan from Tianjin University for the constructive discussion about the application of Propositions 8 and 11. At last, we thank the anonymous reviewers for their detailed reviews, from which we benefit greatly. This work was supported in part by NSF CCF 2212261, NSF IIS 2143493, NSF DMS-2134106, NSF CCF 2019844 and NSF IIS 2110170.

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

# Supplementary Materials

## A   MORE ON RELATED WORKS

In this section, we briefly compare to several related works on sparse linear bandits in terms of regret guarantees, noise assumptions and query models.

- The regret of Abbasi-Yadkori et al. (2012) is $\widetilde{\mathcal{O}}(Rs\sqrt{dT})$ when assuming conditionally $R$-sub-Gaussian noises (i.e., $\eta_t \mid \mathcal{F}_{t-1} \sim \mathrm{subG}(R^2)$, which will be formally defined in **??**). At the same time, they allow an arbitrary varying action set $D_1, D_2, \ldots, D_T \subseteq \mathbb{B}^d$ (though they in fact allows arbitrary decision sets $D_1, D_2, \ldots, D_T \subseteq \mathbb{R}^d$, their regret bound scales with $\max_{x \in D_t} \|x\|_2$, so we assume $D_t \subseteq \mathbb{B}^d$ without loss of generality). This model is less strictive than ours, as we only allow $D_1 = D_2 = \cdots = D_T = \mathbb{B}^d$ (as explained in Footnote 3 in the main text. When the noises are Gaussian with variance 1 and the ground-truth $\theta^*$ is one-hot (i.e., $s = 1$), their regret bound reduces to $\widetilde{\mathcal{O}}(\sqrt{dT})$, which matches the $\Omega(\sqrt{dT})$ bound in Antos & Szepesvári (2009) when the actions sets are allowed to be the entire unit ball (which means the agent will be more powerful than that of Abbasi-Yadkori et al. (2012)).

- The regret of Carpentier & Munos (2012) is $\widetilde{\mathcal{O}}((\|\theta\|_2 + \|\sigma\|_2)s\sqrt{T})$, assuming a unit-ball action set, a $\|\theta^*\|_2 \leq \|\theta\|_2$ ground-truth and $\|\eta_t\|_2 \leq \|\sigma\|_2$ noises, where $\eta_t$ will be defined later. This bound seems to bypass the $\Omega(\sqrt{dT})$ lower bound when $s = 1$. However, this is due to a different noise model: They assumed the noise is component-wise, i.e., $r_t = \langle \theta^* + \eta_t, x_t \rangle$ where $\eta_t \in \mathbb{R}^d$. In contrast, our model assumed a $\langle \theta^*, x_t \rangle + \eta_t$ noise model where $\eta_t \in \mathbb{R}$. Therefore, the $\max_t \|\eta_t\|_2$ dependency can be of order $\mathcal{O}(\sqrt{d})$ to ensure a similar noise model as ours.
- The regret of Lattimore et al. (2015, Appendix G) is also of order $\widetilde{\mathcal{O}}(s\sqrt{T})$, assuming a $[-1, 1]$-bounded noises, a hypercube $\mathcal{X} = [-1, 1]^d$ action set and a $\|\theta^*\|_1 \leq 1$ ground-truth. We will then explain why this does not violate the $\Omega(\sqrt{sdT})$ regret lower bound as well.
  Consider an extreme query $(1, 1, \ldots, 1) \in \mathcal{X}$, which is valid in their query model (in fact, their algorithm is a random projection procedure with some carefully designed regularity conditions, so this type of queries appears all the time). However, in our query model where the action set is the unit ball $\mathbb{B}^d$, we have to scale it by $\frac{1}{\sqrt{d}}$. As the noise will never be scaled, this will amplify the noises by $\sqrt{d}$, so we will need $\text{poly}(d)$ more times of queries to get the same confidence set, making the regret bound have a polynomial dependency on $d$.
  Moreover, their ground-truth $\theta^*$ needs to satisfy $\|\theta^*\|_1 \leq 1$. However, an $s$-sparse ground-truth $\theta^*$ with 2-norm 1 can have 1-norm as much as $\sqrt{s}$. Therefore, another $\sqrt{s}$ should also be multiplied for a fair comparison with our algorithm.

In conclusion, the second and the third work assumed different noise or query models to amplify the signal-to-noise ratio and thus avoid a polynomial dependency on $d$, compared to the regret bounds of Abbasi-Yadkori et al. (2012) and ours.

However, we have to admit that Abbasi-Yadkori et al. (2011) allows a drifting action set, whereas ours only allow a unit-sphere action set, just like Carpentier & Munos (2012). The reason is discussed in Footnote 3 in the main text.

## B  FUTURE DIRECTIONS

First of all, there is still a gap in the worst-case regret in terms of $s$, as the lower bound for sparse linear bandits is $\Omega(\sqrt{sdT})$ instead of our $\widetilde{\mathcal{O}}(s\sqrt{dT})$ when $\sigma_t \equiv 1$. Closing this gap in $s$ is an interesting future work. Our current algorithm, unfortunately, is incapable of a $\mathcal{O}(\sqrt{sdT})$-style worst-case regret guarantee: Suppose that $T = ds^2$, $\theta_i^* = s^{-1/2}$ for $i = 1, 2, \ldots, s$ (so $\|\theta^*\|_2 \leq 1$), and $\sigma_t \equiv 1$. Then we have $n_\Delta^b \approx \Delta^{-1}\sqrt{n_\Delta^b(1 + d\Delta^{-2})}$, which gives $n_\Delta^b \approx d\Delta^{-4}$. Hence, the total regret will be $\sum_{i=1}^s \sum_{\Delta \geq \theta_i^*} n_\Delta^b (\theta_i^*)^2 \approx d\sum_{i=1}^s (\theta_i^*)^2 = ds^2 = \mathcal{O}(s\sqrt{dT})$. Thus, algorithmic improvements must be made to better dependency on $s$. We leave this for future research.

Moreover, the current work relies on the random projection procedure (Carpentier & Munos, 2012), which only works when the action set is the *unit sphere*. Such an assumption is unrealistic in practice. We wonder whether there is an alternative that only requires a looser condition.

At last, deriving a variance-aware lower bound rather than a minimax one is also important, as it can better illustrate the inherent hardness of the problem with different noise levels. We remark that extending the proof of current minimax lower bounds (see, e.g., (Antos & Szepesvári, 2009)) to variance-aware ones is not straightforward.

## C  DIVIDE-AND-CONQUER ALGORITHM FOR DETERMINISTIC SETTINGS

In this section, we discuss how to solve the deterministic sparse linear bandit problem in $\widetilde{\mathcal{O}}(s)$ steps using a divide-and-conquer algorithm, as we briefly mentioned in the main text.

We mainly adopt the idea mentioned by Dong et al. (2021, Footnote 6). For each divide-and-conquer subroutine working on several coordinates $i_1, i_2, \ldots, i_k \in [d]$, we query half of them (e.g., $i_1, i_2, \ldots, i_{k/2}$ when assuming $2 \mid k$) with $\sqrt{2/k}$ mass on each coordinate. This will reveal whether there is a non-zero coordinate among them. If the feedback is non-zero, we then conclude that there exists a non-zero coordinate in this half. Hence, we dive into this half and conquer this sub-problem (i.e., divide-and-conquer). Otherwise, we simply discard this half and consider the other half.

However, this vanilla algorithm proposed by Dong et al. (2021) fails to consider the possibility that two coordinates cancel each other (e.g., two coordinates with magnitude $\pm\sqrt{1/2}$ will make the feedback equal to zero). Fortunately, this problem can be resolved via randomly putting magnitude $\pm\sqrt{2/k}$ on each coordinate, which is similar to the idea illustrated in Algorithm 2. As the environment is deterministic, each step will give the correct feedback with probability 1. Therefore, a constant number of trials is enough to tell whether there exists a non-zero coordinate.

At last, we analyze the number of interactions needed for this approach. As it is guaranteed that each divide-and-conquer subroutine will be working on a set of coordinates where at least one of them is non-zero, we can bound the number of interactions as $\mathcal{O}\left(\sum_{\theta_i^* \neq 0} \#\text{subroutines containing } i\right)$. As for each time we will divide the coordinates into half, there can be at most $\log_2 d$ subroutines containing $i$ for each individual $i$. Therefore, the number of interactions will be $\mathcal{O}(s \log d)$.

After that, we will be sure to find out all coordinates with non-zero magnitudes. Asking each of them once then reveals their actual magnitude. Therefore, we can recover $\theta^*$ in $\mathcal{O}(s \log d + s) = \mathcal{O}(s \log d)$ rounds and will not suffer any regret after that. So the regret of this algorithm will indeed be $\widetilde{\mathcal{O}}(s)$, which is (nearly) independent of $d$ and $T$.

## D    CONCENTRATION INEQUALITIES

### D.1    SAMPLE MEAN UPPER BOUND

We shall make use of the following self-normalizing result.

**Proposition 8** (Fan et al. (2015), Remark 2.9)). *Suppose that $\{\xi_i\}_{i=1}^n$ are independent and symmetric. Then for all $x > 0$,*

$$\Pr\left\{\max_{1 \leq k \leq n} \frac{\sum_{i=1}^k \xi_i}{\sqrt{\sum_{i=1}^n \xi_i^2}} \geq x\right\} \leq \exp\left(-\frac{x^2}{2}\right).$$

**Corollary 9.** *Let $X_1, X_2, \ldots, X_n$ be a sequence of independent and symmetric random variables where $n$ is a stopping time that is finite a.s. Then for any $\delta > 0$, with probability $1 - \delta$, we have*

$$\left|\sum_{i=1}^n (X_i - \mu_i)\right| \leq \sqrt{\sum_{i=1}^n (X_i - \mu_i)^2 \ln \frac{2}{\delta}}.$$

*Proof.* This immediately follows by picking $x = 2\sqrt{\ln \frac{2}{\delta}}$ and then applying Fatou's lemma.    □

Therefore, we can present our Empirical Bernstein Inequality for conditional symmetric stochastic processes with a common mean, as follows:

**Theorem 10** (Empirical Bernstein Inequality). *For a sequence of independent and symmetric random variables $X_1, X_2, \ldots, X_n$ that shares a common mean (i.e., $\mathbb{E}[X_i] = \mu$ for some $\mu$ for all $i$), we have the following inequality where $n$ is a stopping time finite a.s.*

$$\Pr\left\{\left|\sum_{i=1}^n (X_i - \mu)\right| \leq \sqrt{2 \sum_{i=1}^n (X_i - \overline{X})^2 \ln \frac{4}{\delta}}\right\} \geq 1 - \delta, \quad \forall \delta \in (0, 1),$$

*where $\overline{X} = \frac{1}{n} \sum_{i=1}^n X_i$ is the sample mean.*

*Proof.* By direct calculation, we have

$$\sum_{i=1}^n (X_i - \overline{X})^2 = \sum_{i=1}^n X_i^2 - 2n\overline{X}^2 + n\overline{X}^2 = \sum_{i=1}^n X_i^2 - n\overline{X}^2 = \sum_{i=1}^n (X_i - \mu)^2 - n(\overline{X} - \mu)^2.$$

Applying Corollary 9 to $\{X_i\}_{i=1}^n$ gives

$$\Pr\left\{\left|\sum_{i=1}^n (X_i - \mu)\right| \geq \sqrt{\sum_{i=1}^n (X_i - \mu)^2 \ln \frac{4}{\delta}}\right\} \leq \frac{\delta}{2}.$$

Therefore, with probability $1 - \frac{\delta}{2}$, we have

$$\sum_{i=1}^n (X_i - \overline{X})^2 \leq \sum_{i=1}^n (X_i - \mu)^2 \leq \sum_{i=1}^n (X_i - \overline{X})^2 + \frac{4}{n}\sum_{i=1}^n (X_i - \mu)^2 \ln \frac{4}{\delta}. \tag{6}$$

Hence, with probability $1 - \frac{\delta}{2}$, we have

$$\sum_{i=1}^n (X_i - \mu)^2 \leq 2\sum_{i=1}^n (X_i - \overline{X})^2. \tag{7}$$

By the union bound, with probability $1 - \delta$, we thus have

$$\left|\sum_{i=1}^n (X_i - \mu)\right| \leq \sqrt{2\sum_{i=1}^n (X_i - \overline{X})^2 \ln \frac{4}{\delta}},$$

as claimed. $\qquad\square$

## D.2 Sample Variance Upper Bound

Recall that a random variable $X$ is sub-Gaussian with variance proxy $\sigma^2$ if and only if $\mathbb{E}[\exp(\lambda(X - \mathbb{E}[X]))] \leq \exp(\frac{1}{2}\lambda^2\sigma^2)$ for all $\lambda > 0$. We shall denote such a random variable by $X \sim \mathrm{subG}(\sigma^2)$. We first state the following generalized Freedman's inequality for sub-Gaussian random variables.

**Proposition 11** (Fan et al. (2015, Theorem 2.6)). *Suppose that $\{\xi_i\}_{i=1}^n$ is a sequence of zero-mean random variables, i.e., $\mathbb{E}[\xi_i] = 0$. Suppose that $\mathbb{E}[\exp(\lambda\xi_i)] \leq \exp(f(\lambda)V_i)$ for some deterministic function $f(\lambda)$ and some fixed $\{V_i\}_{i=1}^n$ for all $\lambda \in (0,\infty)$, then, for all $x, v > 0$ and $\lambda > 0$, we have*

$$\Pr\left\{\exists 1 \leq k \leq n : \sum_{i=1}^k \xi_i \geq x \wedge \sum_{i=1}^k V_i \leq v^2\right\} \leq \exp\left(-\lambda x + f(\lambda)v^2\right).$$

To derive a bound related to $\sum(X_i - \overline{X})^2$ and $\sum \sigma_i^2$, we will need to characterize the concentration of the square of a sub-Gaussian random variable, which is a "sub-exponential" random variable:

**Proposition 12** (Honorio & Jaakkola (2014, Appendix B)). *For a sub-Gaussian random variable $X$ with variance proxy $\sigma^2$ and mean $\mu$, we have*

$$\mathbb{E}\left[\exp\left(\lambda(X^2 - \mathbb{E}[X^2])\right)\right] \leq \exp\left(16\lambda^2\sigma^4\right), \quad \forall|\lambda| \leq \frac{1}{4\sigma^2}.$$

**Theorem 13.** *For a sequence of sub-Gaussian random variables $\{X_i\}_{i=1}^n$ such that $\mathbb{E}[X_i] = \mu_i$, $X_i \sim \mathrm{subG}(\sigma_i^2)$, and $n$ is a stopping time finite a.s.,*

$$\Pr\left\{\left|\sum_{i=1}^n \left((X_i - \mu_i)^2 - \mathbb{E}[(X_i - \mu_i)^2]\right)\right| > 4\sqrt{2}\sum_{i=1}^n \sigma_i^2 \ln\frac{2}{\delta}\right\} \leq \delta, \quad \forall \delta \in (0,1).$$

*Proof.* We first consider a non-stopping time $n$. Apply Proposition 11 to the sequence $\{(X_i - \mu_i)^2 - \mathbb{E}[(X_i - \mu_i)^2]\}$ with $V_i = \sigma_i^4$, $f(\lambda) = 16\lambda^2$ for $\lambda < \frac{1}{4\sigma_{\max}^2}$ and $f(\lambda) = \infty$ otherwise, where $\sigma_{\max}$ is defined as $\max\{\sigma_1, \sigma_2, \ldots, \sigma_n\}$. Then for all $x, v > 0$ and $\lambda \in (0, \frac{1}{\sigma_{\max}^2})$, we have

$$\Pr\left\{\sum_{i=1}^n \left((X_i - \mu_i)^2 - \mathbb{E}[(X_i - \mu_i)^2]\right) > x \wedge \sqrt{\sum_{i=1}^n \sigma_i^4} \leq v\right\} \leq \exp\left(-\lambda x + 16\lambda^2 v^2\right).$$

Picking $v^2 = \sum_{i=1}^n \sigma_i^4 \sqrt{\ln \frac{2}{\delta}}$ and $x = 4\sqrt{2}v \ln \frac{2}{\delta}$ gives

$$\Pr \left\{ \sum_{i=1}^n \left( (X_i - \mu_i)^2 - \mathbb{E}[(X_i - \mu_i)^2] \right) > 4\sqrt{2 \sum_{i=1}^n \sigma_i^4 \ln \frac{2}{\delta}} \right\} \leq \exp \left( -\frac{x^2}{32v^2} \right)$$

$$= \exp \left( -\frac{32v^2 \ln^2 \frac{2}{\delta}}{32v^2} \right) = \frac{\delta}{2},$$

where $\lambda$ is set to $\frac{x}{32v^2} = \frac{1}{4\sqrt{2}v} < \frac{1}{\sigma_{\max}^2}$ (as $v^2 > \sum_{i=1}^n \sigma_i^4 \geq \sigma_{\max}^4$). A union bound by applying Proposition 11 to the sequence $\{\mathbb{E}[(X_i - \mu_i)^2] - (X_i - \mu_i)^2\}$ with the same parameters and noticing the fact that $\sqrt{\sum_{i=1}^n \sigma_i^4} \leq \sum_{i=1}^n \sigma_i^2$ then shows that our conclusion hold for any fixed $n$. By Fatou's lemma, we conclude that it also holds for a stopping time $n$ that is finite a.s. $\qquad\square$

**Theorem 14** (Variance Concentration). *Let $\{X_i\}_{i=1}^n$ be a sequence of random variables with a common mean $\mu$ such that $X_i \sim subG(\sigma_i^2)$, $(X_i - \mu)$ is symmetric, and $n$ is a stopping time finite a.s. Then, $\forall \delta \in (0,1)$, with probability $1 - \delta$, we have the following three inequalities:*

$$\sum_{i=1}^n (X_i - \overline{X})^2 \leq \sum_{i=1}^n (X_i - \mu)^2$$

$$\sum_{i=1}^n (X_i - \overline{X})^2 \geq \frac{1}{2} \sum_{i=1}^n (X_i - \mu)^2$$

$$\sum_{i=1}^n (X_i - \mu)^2 \leq 8 \sum_{i=1}^n \sigma_i^2 \ln \frac{2}{\delta}.$$

*where $\overline{X} = \frac{1}{n} \sum_{i=1}^n X_i$ is the sample mean.*

*Proof.* The first two inequalities follow from Eqs. (6) and (7). The last one follows from Theorem 13 together with the fact that $\mathbb{E}[(X_i - \mu_i)^2] \leq \sigma_i^2$ by definition of sub-Gaussian random variables. $\quad\square$

## E    RIDGE LINEAR REGRESSION

**Lemma 15.** *Suppose that we are given $n$ samples $y_i = \langle x_i, \beta^* \rangle + \epsilon_i$, $i = 1, 2, \ldots, n$, where $\beta^* \in \mathbb{B}^d$ and $\{x_i\}_{i=1}^n, \{\epsilon_i\}_{i=1}^n$ are stochastic processes adapted to the filtration $\{\mathcal{F}_i\}_{i=0}^n$ such that $\epsilon_i$ is conditionally $\overline{\sigma}$-Gaussian, i.e., $\epsilon_i \mid \mathcal{F}_{i-1} \sim subG(\overline{\sigma}^2)$. Define the following quantity as the estimate for $\beta^*$:*

$$\widehat{\beta} = \underset{\beta}{\arg\min} \left( \sum_{i=1}^n (y_i - x_i^\mathsf{T} \beta)^2 + \lambda \|\beta\|_2 \right).$$

*Then with probability $1 - \delta$, the following inequality holds:*

$$\left| \sum_{i=1}^n (y_i - x_i^\mathsf{T} \beta^*)^2 - \sum_{i=1}^n (y_i - x_i^\mathsf{T} \widehat{\beta})^2 \right| \leq 2d\overline{\sigma}^2 \ln \frac{n}{\lambda d \delta^2} + \frac{1}{\lambda} + \lambda.$$

*Proof.* Denote $y = (y_1, y_2, \ldots, y_n)^\mathsf{T}$, $X = (X_1^\mathsf{T}, X_2^\mathsf{T}, \ldots, X_n^\mathsf{T})$ and $\epsilon = (\epsilon_1, \epsilon_2, \ldots, \epsilon_n)^\mathsf{T}$. Denote $\mathrm{Var}^* = \sum_{i=1}^n (y_i - x_i^\mathsf{T} \beta^*)^2$ and $\widehat{\mathrm{Var}} = \sum_{i=1}^n (y_i - x_i^\mathsf{T} \widehat{\beta})^2$. We have the following representation of $\widehat{\beta}$, which is by direct calculation (check, e.g., (Kirschner & Krause, 2018))

$$\widehat{\beta} = (X^\mathsf{T} X + \lambda I)^{-1} X^\mathsf{T} y. \tag{8}$$

Furthermore, by Abbasi-Yadkori et al. (2011, Proof of Theorem 2), we have

$$\widehat{\beta} - \beta^* = (X^\mathsf{T} X + \lambda I)^{-1} X^\mathsf{T} \epsilon - \lambda (X^\mathsf{T} X + \lambda I)^{-1} \beta^*. \tag{9}$$

Therefore, we can write

$$
\begin{aligned}
\mathrm{Var}^* - \widehat{\mathrm{Var}} &= (y - X\beta^*)^\mathsf{T}(y - X\beta^*) - (y - X\widehat{\beta})^\mathsf{T}(y - X\widehat{\beta}) \\
&= (\beta^*)^\mathsf{T} X^\mathsf{T} X \beta^* - (\beta^*)^\mathsf{T} X^\mathsf{T} y - y^\mathsf{T} X \beta^* - \widehat{\beta}^\mathsf{T} X^\mathsf{T} X \widehat{\beta} + \widehat{\beta}^\mathsf{T} X^\mathsf{T} y + y^\mathsf{T} X \widehat{\beta} \\
&= \widehat{\beta}^\mathsf{T}(X^\mathsf{T} y - X^\mathsf{T} X \widehat{\beta}) - (\beta^*)^\mathsf{T} X^\mathsf{T}(y - X\beta^*) + y^\mathsf{T} X(\widehat{\beta} - \beta^*).
\end{aligned}
$$

By Eq. (8), we have $X^\mathsf{T} y = (X^\mathsf{T} X + \lambda I)\widehat{\beta}$. So the first term is just $\lambda \widehat{\beta}^\mathsf{T} \widehat{\beta}$. As $y = X\beta^* + \epsilon$, the second term is just $-(\beta^*)^\mathsf{T} X^\mathsf{T} \epsilon$. By Eq. (9), the last term becomes

$$
y^\mathsf{T} X(X^\mathsf{T} X + \lambda I)^{-1} X^\mathsf{T} \epsilon - \lambda y^\mathsf{T} X(X^\mathsf{T} X + \lambda I)^{-1}\beta^*.
$$

For the sake of simplicity, we define $\langle a, b \rangle_M = a^\mathsf{T}(X^\mathsf{T} X + \lambda I)^{-1} b$ (note that $\langle a, b \rangle_M = \langle b, a \rangle_M$ as $X^\mathsf{T} X + \lambda I$ is symmetric) and denote the induced norm by $\|\cdot\|_M$. Therefore, we have

$$
\mathrm{Var}^* - \widehat{\mathrm{Var}} = \lambda \widehat{\beta}^\mathsf{T} \widehat{\beta} - (\beta^*)^\mathsf{T} X^\mathsf{T} \epsilon + \langle X^\mathsf{T} y, X^\mathsf{T} \epsilon \rangle_M - \lambda \langle X^\mathsf{T} y, \beta^* \rangle_M.
$$

Again by Eq. (8), we have $\langle X^\mathsf{T} y, X^\mathsf{T} \epsilon \rangle_M = \widehat{\beta}^\mathsf{T} X^\mathsf{T} \epsilon$. Therefore, the second and third term together give

$$
\begin{aligned}
&- (\beta^*)^\mathsf{T} X^\mathsf{T} \epsilon + \langle X^\mathsf{T} y, X^\mathsf{T} \epsilon \rangle_M = (\widehat{\beta} - \beta^*)^\mathsf{T} X^\mathsf{T} \epsilon \\
&= \left((X^\mathsf{T} X + \lambda I)^{-1} X^\mathsf{T} \epsilon - \lambda(X^\mathsf{T} X + \lambda I)^{-1}\beta^*\right)^\mathsf{T} X^\mathsf{T} \epsilon \\
&= \|X^\mathsf{T} \epsilon\|_M^2 - \lambda \langle \beta^*, X^\mathsf{T} \epsilon \rangle = \|X^\mathsf{T} \epsilon\|_M^2 - \lambda(\beta^*)^\mathsf{T}(\widehat{\beta} - \beta^*) + \lambda^2 \langle \beta^*, \beta^* \rangle_M,
\end{aligned}
$$

where the last step is yielded from using Eq. (9) reversely. Then note that $\langle X^\mathsf{T} y, \beta^* \rangle = \widehat{\beta}^* \beta^*$, so taking expectation on both sides gives

$$
\begin{aligned}
\left| \mathrm{Var}^* - \widehat{\mathrm{Var}} \right| &= \lambda \widehat{\beta}^\mathsf{T} \widehat{\beta} + \|X^\mathsf{T} \epsilon\|_M^2 - \lambda(\beta^*)^\mathsf{T}(\widehat{\beta} - \beta^*) + \lambda^2 \langle \beta^*, \beta^* \rangle_M - \lambda \widehat{\beta}^\mathsf{T} \beta^* \\
&= \|X^T \epsilon\|_M^2 + \lambda \|\widehat{\beta} - \beta^*\|_2^2 + \lambda^2 \|\beta^*\|_M^2.
\end{aligned}
$$

By Cauchy-Schwartz inequality, we have

$$
\|\widehat{\beta} - \beta^*\|_2^2 \le \|X^\mathsf{T} \epsilon\|_M^2 \|(X^\mathsf{T} X + \lambda I)^{-1/2}\|^2 + \|\beta^*\|_M^2 \|(X^\mathsf{T} X + \lambda I)^{-1/2}\|^2,
$$

where the matrix norms can further be bounded by $1/\lambda_{\min}(X^\mathsf{T} X + \lambda I) \le \frac{1}{\lambda}$. Similarly, we can conclude that $\|\beta^*\|_M^2 \le 1/\lambda_{\min}(X^\mathsf{T} X + \lambda I)\|\beta^*\|_2^2 \le 1/\lambda$. Consequently, we have

$$
\left| \mathrm{Var}^* - \widehat{\mathrm{Var}} \right| = 2\|X^T \epsilon\|_M^2 + \frac{1}{\lambda} + \lambda.
$$

As proved by Abbasi-Yadkori et al. (2011, Theorem 1), with probability $1 - \delta$, we have

$$
\|X^\mathsf{T} \epsilon\|_M^2 \le 2\overline{\sigma}^2 \ln\left( \frac{1}{\delta} \frac{\sqrt{\det(X^\mathsf{T} X + \lambda I)}}{\sqrt{\det(\lambda I)}} \right) = \overline{\sigma}^2 \ln\left( \frac{1}{\delta^2} \frac{(\lambda + n/s)^s}{\lambda} \right) \le d\overline{\sigma}^2 \ln\left( \frac{1}{\delta^2}\left(1 + \frac{n}{\lambda d}\right) \right),
$$

where $\overline{\sigma}^2$ is the (maximum) variance proxy and the second last step is due to the Determinant-Trace Inequality (Abbasi-Yadkori et al., 2011, Lemma 10). Hence, with probability $1 - \delta$, we have

$$
\left| \mathrm{Var}^* - \widehat{\mathrm{Var}} \right| \le 2d\overline{\sigma}^2 \ln \frac{n}{\lambda d \delta^2} + \frac{1}{\lambda} + \lambda,
$$

as claimed. □

# F    OMITTED PROOF IN SECTION 3.2 (ANALYSIS OF FRAMEWORK)

## F.1    PROOF OF MAIN THEOREM

In this section, we prove Theorem 3, which is restated as follows for the ease of reading:

**Theorem 16** (Restatement of Theorem 3). *Suppose that for any execution of $\mathcal{F}$ that last for $n$ steps, $\overline{\mathcal{R}_n^{\mathcal{F}}} \geq \mathcal{R}_n^{\mathcal{F}}$ holds with probability $1 - \delta$, i.e., $\overline{\mathcal{R}_n^{\mathcal{F}}}$ is pessimistic. Then the total regret incurred by the second phase satisfies*

$$\mathcal{R}_T^b = \widetilde{\mathcal{O}}\left( s\sqrt{d}\sqrt{\sum_{t=1}^{T} \sigma_t^2 \log \frac{1}{\delta}} + s \log \frac{1}{\delta} \right) \quad \text{with probability } 1 - \delta.$$

*Proof.* As mentioned in the main body, we define the following good events for each $\Delta = 2^{-2}, 2^{-2}, \ldots, \Delta_f$ where $\Delta_f$ is the final value of $\Delta$:

$$\mathcal{G}_\Delta : \sum_{i \in S} \theta_i^*(\theta_i^* - \widehat{\theta}_i) < \Delta^2 \text{ after } \mathcal{T}_\Delta^a,$$

$$\mathcal{H}_\Delta : (|\theta_i^*| > 3\Delta \to i \in S) \wedge \left( i \in S \to |\theta_i^*| > \frac{1}{2}\Delta \right) \text{ after } \mathcal{T}_\Delta^b. \tag{10}$$

It is by definition to see that $\mathcal{H}_{\frac{1}{2}}$ indeed holds, as all $|\theta_i^*| < \frac{3}{2}$ and $S$ is initially empty. We can then use induction to prove that all good events hold with high probability.

Here, we list several technical lemmas we informally referred to in the main body, whose proofs are left to subsequent sections. The first one is about the regret-to-sample-complexity conversion.

**Lemma 17** (Regret-to-sample-complexity Conversion). *If for any execution of $\mathcal{F}$ that lasts for $n$ steps, we have $\overline{\mathcal{R}_n^{\mathcal{F}}} \geq \mathcal{R}_n^{\mathcal{F}}$ with probability $1 - \delta$, then we have $\Pr\{\mathcal{G}_\Delta\} \geq 1 - \delta$.*

The second term bounds the 'bias' term, i.e., the second term of Eq. (4) for random projection.

**Lemma 18** (Bias Term of the Random Projection). *Conditioning on $\mathcal{G}_\Delta$ and $\mathcal{H}_{2\Delta}$, we have $|\sum_{i \in S}(\widehat{\theta}_i^2 - (\theta_i^*)^2)| < 3\Delta^2$, which further gives $|\sum_{i \in S} \widehat{\theta}_i(\theta_i^* - \widehat{\theta}_i)| < 4\Delta^2$.*

Furthermore, we can bound the estimation error of the random projection process.

**Lemma 19** (Concentration of the Random Projection). *For any given $i \in [K]$, we will have*

$$\Pr\left\{ |\bar{r}_i - \theta_i^*| > 3\Delta^2 + \frac{\Delta}{4} \middle| \mathcal{G}_\Delta, \mathcal{H}_{2\Delta} \right\} \leq \delta.$$

As long as the estimation errors are small, we can ensure, with high probability that the good event for $\Delta$, namely $\mathcal{H}_\Delta$ will also hold.

**Lemma 20** (Identification of Non-zero Coordinates). $\Pr\{\mathcal{H}_\Delta \mid \mathcal{G}_\Delta, H_{2\Delta}\} \geq 1 - d\delta.$

Therefore, by combining all lemmas above, we can ensure that all good events, namely $\{\mathcal{G}_\Delta \wedge \mathcal{H}_\Delta\}_{\Delta = 2^{-2}, \ldots, \Delta_f}$, hold simultaneously with probability $1 - dT\delta$.

At last, we bound the total regret incurred in Phase B for $\Delta$.

**Lemma 21** (Single-Phase Regret Bound). *Conditioning on $\mathcal{G}_\Delta$ and $\mathcal{H}_{2\Delta}$, we have*

$$\sum_{t \in \mathcal{T}_\Delta^b} \langle \theta^* - x_t, \theta^* \rangle \leq 36sn_\Delta^b \Delta^2 + 6s\Delta \frac{R}{\sqrt{K}} \sqrt{2n_\Delta^b \ln \frac{1}{\delta}}, \quad \text{with probability } 1 - s\delta.$$

Then we follow the analysis sketched in the main body. We assume, without loss of generality, that $s < d$. Then, conditioning on all $\mathcal{G}_\Delta$ and $\mathcal{H}_\Delta$, some coordinate $i^*$ must never be included into $S$ as $\mathcal{H}_\Delta$ holds for all $\Delta$.

Therefore, by property of the sample variances (Theorem 14), for such $i^*$ and for each phase, with probability $1 - \delta$, we have $\sum_{n=1}^N (r_{n,i^*} - \bar{r}_{i^*})^2 \leq \sum_{n=1}^N (r_{n,i^*} - \mathbb{E}[r_{n,i^*}])^2$. Together with Eq. (2),

$$2\sqrt{2\sum_{t\in\widetilde{\mathcal{T}}_\Delta^b}(r_{t,i^*}-\mathbb{E}[r_{t,i^*}])^2\ln\frac{4}{\delta}} > (n_\Delta^b-1)\frac{\Delta}{4}. \tag{11}$$

By using Lemma 21, the total regret from Phase B is bounded by

$$\mathcal{R}_T^b \leq \sum_{\Delta=2^0,2^{-2},\ldots,\Delta_f}\left(36sn_\Delta^b\Delta^2 + 6s\Delta\frac{R}{\sqrt{K}}\sqrt{2n_\Delta^b\ln\frac{1}{\delta}}\right)$$

By writing $n_\Delta^b$ as $(n_\Delta^b - 1) + 1$, for any given $\Delta$, the total regret of Phase b is bounded by

$$\mathcal{R}_T^b \leq \sum_\Delta\sum_{t\in\mathcal{T}_\Delta^b}\langle\theta^* - x_t,\theta^*\rangle$$

$$\leq \sum_\Delta 36s\Delta^2\cdot\left(\frac{4}{\Delta}2\sqrt{2\sum_{t\in\widetilde{\mathcal{T}}_\Delta^b}(r_{t,i^*}-\mathbb{E}[r_{t,i^*}])^2\ln\frac{4}{\delta}}+1\right) +$$

$$\sum_\Delta 6s\Delta\frac{R}{\sqrt{K}}\sqrt{2\ln\frac{1}{\delta}}\sqrt{4\Delta\cdot2\sqrt{2\sum_{t\in\widetilde{\mathcal{T}}_\Delta^b}(r_{t,i^*}-\mathbb{E}[r_{t,i^*}])^2\ln\frac{4}{\delta}}+1}$$

$$\leq \underbrace{\sum_\Delta\left(288\sqrt{2}s\sqrt{\Delta^2\sum_{t\in\mathcal{T}_\Delta^b}(r_{t,i^*}-\mathbb{E}[r_{t,i^*}])^2\ln\frac{4}{\delta}}+36s\Delta^2\right)}_{\text{Part (a)}} + \tag{12}$$

$$\underbrace{\sum_\Delta\left(24s\sqrt{\ln\frac{1}{\delta}}\Delta\sqrt[4]{\Delta^2\sum_{t\in\mathcal{T}_\Delta^b}(r_{t,i^*}-\mathbb{E}[r_{t,i^*}])^2\ln\frac{4}{\delta}}+12s\Delta\sqrt{\ln\frac{1}{\delta}}\right)}_{\text{Part (b)}}. \tag{13}$$

As mentioned in the main text, we make use of the following lower bound of $n_\Delta^b$ which again follows from Eq. (2) and holds for all $\Delta$'s except $\Delta_f$:

$$n_\Delta^b \geq \frac{4}{\Delta}\cdot2\sqrt{2\sum_{t\in\mathcal{T}_\Delta^b}(r_{t,i^*}-\bar{r}_{i^*})^2\ln\frac{4}{\delta}} \geq \frac{8}{\Delta}\sqrt{2\sum_{t\in\mathcal{T}_\Delta^b}(r_{t,i^*}-\mathbb{E}[r_{t,i^*}])^2\ln\frac{4}{\delta}},$$

where the last step is due to Theorem 14. For simplicity, define $S_\Delta = \Delta^2\sum_{t\in\mathcal{T}_\Delta^b}(r_{n,i}-\mathbb{E}[r_{n,i}])^2$ and therefore $n_\Delta^b \geq \frac{C}{\Delta^2}\sqrt{S_\Delta}$ where $C = 8\sqrt{2\ln\frac{4}{\delta}}$, a constant if we regard $\delta$ as a constant. Therefore,

$$\sum_{\Delta\neq\Delta_f}\frac{C}{\Delta^2}\sqrt{S_\Delta} \leq \sum_{\Delta\neq\Delta_f}n_\Delta^b \leq T \tag{14}$$

and our goal is to upper bound $\sum_\Delta\sqrt{S_\Delta}$. Define a threshold $X = T/\left(C\sqrt{\sum_{\Delta\neq\Delta_f}S_\Delta}\right)$ and denote $\Delta_X = 2^{-\lceil\log_4 X\rceil}$ so that $\Delta_X^2 \leq \frac{1}{X}$. We will have

$$\sum_{\Delta\neq\Delta_f}\sqrt{S_\Delta} = \sum_{\Delta=2^{-2},\ldots,\Delta_X}\sqrt{S_\Delta} + \sum_{\Delta=\Delta_X/2,\ldots,2\Delta_f}\sqrt{S_\Delta}$$

$$\overset{(a)}{\leq} \sqrt{\log_4 X} \sqrt{\sum_{\Delta = 2^{-2}, \dots, \Delta_X} S_\Delta} + \sum_{\Delta_X > \Delta > \Delta_f} \frac{\Delta_X^2}{\Delta_X^2} \sqrt{S_\Delta}$$

$$\overset{(b)}{\leq} \sqrt{\log_4 X} \sqrt{\sum_{\frac{1}{4} \geq \Delta \geq \Delta_X} S_\Delta} + \frac{1}{X} \sum_{\Delta_X > \Delta > \Delta_f} \frac{\sqrt{S_\Delta}}{\Delta_X^2}$$

$$\overset{(c)}{\leq} \sqrt{\log_4 X} \sqrt{\sum_{\frac{1}{4} \geq \Delta \geq \Delta_X} S_\Delta} + \frac{1}{X} \frac{T}{C} = (\sqrt{\log_4 X} + 1) \sqrt{\sum_{\Delta \neq \Delta_f} S_\Delta}, \qquad (15)$$

where (a) applied Cauchy-Schwartz to the first summation, (b) applied the fact that $\Delta_X^2 \leq \frac{1}{X}$ and (c) used Eq. (14). So Part (a) of the regret, namely Eq. (12), can be bounded by

$$\sum_\Delta \left( 288\sqrt{2}s \sqrt{\Delta^2 \sum_{t \in \mathcal{T}_\Delta^b} (r_{t,i} - \mathbb{E}[r_{t,i}])^2 \ln \frac{4}{\delta} + 36s\Delta^2} \right)$$

$$\overset{(a)}{\leq} 288\sqrt{2}s \sum_\Delta \sqrt{S_\Delta \ln \frac{4}{\delta}} + 72s \overset{(b)}{\leq} 288\sqrt{2}s \sqrt{\ln \frac{4}{\delta}} \left( \sqrt{\log_4(4T) \sum_{\Delta \neq \Delta_f} S_\Delta} + \sqrt{S_{\Delta_f}} \right) + 72s$$

$$\overset{(c)}{\leq} 576s \sqrt{\sum_{\Delta = 2^{-2}, \dots, \Delta_f} \Delta^2 \sum_{t \in \mathcal{T}_\Delta^b} (r_{t,i^*} - \mathbb{E}[r_{t,i^*}])^2 \ln \frac{4}{\delta} \log_4(4T)} + 72s$$

$$\overset{(d)}{\leq} 576s \sqrt{\sum_{\Delta = 2^{-2}, \dots, \Delta_f} 8 \sum_{t \in \mathcal{T}_\Delta^b} (\Delta^2 + d\sigma_t^2) \ln \frac{4}{\delta} \log_4(4T)} + 72s \qquad (16)$$

$$\leq 1152\sqrt{2}s\sqrt{d}\sqrt{\sum_{t=1}^T \sigma_t^2 \ln \frac{4}{\delta} \log_4(4T)} + 1152\sqrt{2}s \sqrt{\sum_\Delta n_\Delta^b \Delta^2 \ln \frac{1}{\delta} \log_4(4T)} + 72s.$$

where (a) used the fact that $\sum \Delta^2 \leq 2$ as $\Delta = 2^{-i}$, (b) used Eq. (15), (c) again used Cauchy-Schwartz inequality and (d) used Theorem 14(3), the variance concentration result, Eq. (19), the magnitude of the variance, and the facts that $R \geq \Delta$, $K \leq d$. Therefore, we can conclude that

$$\mathcal{O}(s) \cdot \sum_\Delta n_\Delta^b \Delta^2 \leq \tilde{\mathcal{O}} \left( s\sqrt{d}\sqrt{\sum_{t=1}^T \sigma_t^2 \log \frac{4}{\delta}} + s \right) + \tilde{\mathcal{O}} \left( s\sqrt{n_\Delta^b \Delta^2 \ln \frac{1}{\delta}} \right).$$

Notice that the left-handed-side and the right-handed-side has a common term (the RHS one is inside the square-root sign). Hence, by the self-bounding property Lemma 37, we can conclude that (note that we divided $s$ on both sides)

$$\sum_\Delta n_\Delta^b \Delta^2 \leq \tilde{\mathcal{O}} \left( \sqrt{d}\sqrt{\sum_{t=1}^T \sigma_t^2 \log \frac{4}{\delta}} + 1 \right) + \tilde{\mathcal{O}} \left( \ln \frac{1}{\delta} \right), \qquad (17)$$

which means that Part (a) (Eq. (12)), or equivalently $\mathcal{O}(s \sum_\Delta n_\Delta^b \Delta^2)$, is bounded by

$$\tilde{\mathcal{O}} \left( s\sqrt{d}\sqrt{\sum_{t=1}^T \sigma_t^2 \log \frac{4}{\delta}} + s \log \frac{4}{\delta} \right).$$

Now consider Part (b) of the regret, namely Eq. (13). The second term in each summand will sum up to $\mathcal{O}(s\sqrt{\log \frac{1}{\delta}})$. For the first term, using the notation $S_\Delta$, we want to bound

$$24s \left( \ln \frac{1}{\delta} \right)^{3/4} \sum_\Delta \Delta \sqrt[4]{S_\Delta} \leq 24s \ln \frac{1}{\delta} \sum_\Delta \Delta \sqrt[4]{\sum_\Delta S_\Delta} \leq 48s \ln \frac{1}{\delta} \sqrt[4]{\sum_\Delta S_\Delta}.$$

Conditioning on the same events as in Eq. (16), we will have

$$\sum_\Delta S_\Delta \leq \mathcal{O}\left(\sum_\Delta n_\Delta^b \Delta^2 + \sum_{t=1}^T d\sigma_t^2\right) \leq \widetilde{\mathcal{O}}\left(\sum_{t=1}^T d\sigma_t^2 + \sqrt{\sum_{t=1}^T d\sigma_t^2 \log\frac{1}{\delta}} + \log\frac{1}{\delta}\right) \leq \widetilde{\mathcal{O}}\left(\sum_{t=1}^T d\sigma_t^2 \log\frac{1}{\delta}\right),$$

where the second step comes from Eq. (17). Therefore,

$$24s\left(\ln\frac{1}{\delta}\right)^{3/4}\sum_\Delta \Delta\sqrt[4]{S_\Delta} \leq \widetilde{\mathcal{O}}\left(s\sqrt{\sqrt{d}\sqrt{\sum_{t=1}^T \sigma_t^2 \log\frac{1}{\delta}}\left(\log\frac{1}{\delta}\right)^{3/4}}\right) = \widetilde{\mathcal{O}}\left(s\sqrt[4]{d\sum_{t=1}^T \sigma_t^2 \log\frac{1}{\delta}}\right),$$

which gives (by combining the bound of Eq. (12) and Eq. (13) together):

$$\mathcal{R}_T^b \leq \widetilde{\mathcal{O}}\left(s\sqrt{d}\sqrt{\sum_{t=1}^T \sigma_t^2 \log\frac{4}{\delta}} + s\sqrt[4]{d\sum_{t=1}^T \sigma_t^2 \log\frac{1}{\delta}} + s\log\frac{4}{\delta}\right).$$

Further notice that, if $d\sum_{t=1}^T \sigma_t^2 \geq 1$, then the square-root is larger than the 4th-root. When $d\sum_{t=1}^T \sigma_t^2 \leq 1$, then either root is bounded by $s$. Hence, in either case, the 4th-root will be hidden by other factors. Henceforth, we indeed have the following conclusion with probability $1 - (s + 3T)\delta$:

$$\mathcal{R}_T^b \leq \widetilde{\mathcal{O}}\left(s\sqrt{d\sum_{t=1}^T \sigma_t^2 \log\frac{1}{\delta}} + s\log\frac{1}{\delta}\right).$$

By setting the actual $\delta$ as $(s + 3T)\delta$, we will still have the same regret bound as $\mathcal{O}(\log\frac{c}{\delta}) = \mathcal{O}(\log\frac{1}{\delta} + \log c) = \widetilde{\mathcal{O}}(\log\frac{1}{\delta})$ as all logarithmic factors will be hidden by $\widetilde{\mathcal{O}}$. □

### F.2 REGRET-TO-SAMPLE-COMPLEXITY CONVERSION

*Proof of Lemma 17.* By Fatou's lemma and the fact that $n_\Delta^b$ is finite a.s. (as it is truncated according to $T$), the probability that $\overline{\mathcal{R}_n^\mathcal{F}}$ is a pessimistic estimation for all $n = 1, 2, \ldots, n_\Delta^b$ is bounded by $1 - \delta$. Conditioning on this, by definition, we will have

$$\sum_{n=1}^{n_\Delta^b} \langle \theta_i^*, \theta_i^* - x_n \rangle \leq \mathcal{R}_{n_\Delta^b}^\mathcal{F} \leq \overline{\mathcal{R}_{n_\Delta^b}^\mathcal{F}},$$

which means our stopping criterion (Line 5) will ensure

$$\sum_{n=1}^{n_\Delta^b} \langle \theta_i^*, \theta_i^* - \widehat{\theta} \rangle \leq \frac{\overline{\mathcal{R}_{n_\Delta^b}^\mathcal{F}}}{n_\Delta^b} < \Delta^2$$

and thus $\mathcal{G}_\Delta$ is ensured. □

### F.3 RANDOM PROJECTION

*Proof of Lemma 18.* By $\mathcal{H}_{2\Delta}$, we have $|\theta_i^*| > \Delta, \forall i \in S$, which gives $|\theta_i^* - \widehat{\theta}_i| < \Delta$ by $\mathcal{G}_\Delta$. So we can bound $|\sum_{i \in S}(\widehat{\theta}_i^2 - (\theta_i^*)^2)|$ by $2|\sum_{i \in S}\theta_i^*(\theta_i^* - \widehat{\theta}_i)| + |\sum_{i \in S}(\theta_i^* - \widehat{\theta}_i)^2| < 2\Delta^2 + \Delta^2 = 3\Delta^2$. The second claim is then straightforward as $\widehat{\theta}_i(\theta_i^* - \widehat{\theta}_i) = -\theta_i^*(\theta_i^* - \widehat{\theta}_i) - \left((\widehat{\theta}_i^2 - (\theta_i^*)^2)\right)$. □

*Proof of Lemma 19.* In the random projection procedure, let $r_i^{(n)}$ be the random variable defined as (which is the $i$-th coordinate of $r_n$ in the algorithm)

$$
\begin{aligned}
r_i^{(n)} &= \frac{K}{R^2} y_i^{(n)} \left( \langle x_n, \theta^* \rangle + \eta_n - \sum_{j \in S} \widehat{\theta}_j^2 \right) \\
&= \frac{K}{R^2} \left( y_i^{(n)} \right)^2 \theta_i^* + \sum_{j \in S} \widehat{\theta}_j (\theta_j^* - \widehat{\theta}_j) + \sum_{j \notin S, j \neq i} \left( \frac{K}{R^2} y_i^{(n)} y_j^{(n)} \right) \theta_j^* + \left( \frac{K}{R^2} y_i^{(n)} \right) \eta_n. \quad (18)
\end{aligned}
$$

Then $\overline{r}_i$ is just the sample mean estimator of $\{r_i^{(n)}\}_{n=1}^{n_\Delta^b}$.

Firstly, observe that $y_i^2$ is always $\frac{R^2}{K}$, so the first term of Eq. (18) is just $\theta_i^*$, which is exactly the magnitude we want to estimate. Moreover, by Lemma 18, the second term is a small (deterministic) bias added to $\theta_i^*$ and is bounded by $3\Delta^2$. For the third term, as each $y_j$ is independent, $\frac{K}{R^2} y_i y_j$ is an i.i.d. Rademacher random variable, denoted by $z_j^{(n)}$. Hence, the last two terms sum to

$$
\sum_{j \notin S, j \neq i} z_j^{(n)} \theta_j^* + \left( \frac{K}{R^2} y_i^{(n)} \right) \eta_n.
$$

By definition of Rademacher random variables, they are all zero-mean and symmetric. Henceforth they can be viewed as noises with variances at most

$$
\mathbb{E} \left[ \left( \sum_{j \notin S, j \neq i} z_j^{(n)} \theta_j^* + \frac{K}{R^2} y_i^{(n)} \eta_n \right)^2 \right] \overset{(a)}{=} \mathbb{E} \left[ \sum_{j \notin S} \left( z_j^{(n)} \theta_j^* \right)^2 + \left( \frac{K}{R^2} y_i^{(n)} \right)^2 \eta_n^2 \right] \leq 1 + \frac{K}{R^2} \sigma_n^2, \tag{19}
$$

where (a) is due to the mutual independence between $z_j^{(n)}$ and $y_i^{(n)}$.

Moreover, as each $y_i^{(n)}$ is also redrawn for every $n$ and $i$, we know that all $\{z_j^{(n)}\}_{j \notin S, 1 \leq n \leq n_\Delta^b}$ and $\{y_i^{(n)}\}_{1 \leq n \leq n_\Delta^b}$ are all mutually independent. Because the first two terms Eq. (18) is not random, all $\{r_i^{(n)}\}_{1 \leq n \leq n_\Delta^b}$ are independent, symmetric and sub-Gaussian (as $\theta_j^*$ is bounded and $\eta_n$ is Gaussian) random variables. Therefore, as $n_\Delta^b$ is indeed a stopping time finite a.s., we shall apply Empirical Bernstein Inequality (Theorem 10) to $\{r_i^{(n)}\}_n$ where $\mathrm{Var}(r_i^{(n)})$ is characterized by Eq. (19), giving

$$
\Pr \left\{ \left| \sum_{n=1}^{n_\Delta^b} \left( \sum_{j \notin S, j \neq i} z_j^{(n)} \theta_j^* + \left( \frac{K}{R^2} y_i^{(n)} \right) \eta_n \right) \right| \geq 2 \sqrt{2 \sum_{n=1}^{n_\Delta^b} (r_{n,i} - \overline{r}_i)^2 \ln \frac{4}{\delta}} \right\} \leq \delta.
$$

In other words, our choice of $n_\Delta^b$ in Eq. (2) will ensure the average noise is bounded by $\frac{\Delta}{2}$. By Lemma 18, we conclude that $\Pr\{|\overline{r}_i - \theta_i^*| > 3\Delta^2 + \frac{\Delta}{4} \mid \mathcal{G}_\Delta, \mathcal{H}_{2\Delta}\} \leq \delta$. □

*Proof of Lemma 20.* If we skipped due to $\sum_{i \in S} \widehat{\theta}_i^2 > 1 - \Delta^2$, which is Line 7 of Algorithm 1, then by Lemma 18, we will have

$$
\sum_{i \in S} (\theta_i^*)^2 > 1 - 5\Delta^2 \quad \text{conditioning on } \mathcal{G}_\Delta \wedge \mathcal{H}_{2\Delta},
$$

and thus all remaining coordinates are smaller than $3\Delta$. Moreover, by $\mathcal{H}_{2\Delta}$, all discovered coordinates are with magnitude at least $\frac{2\Delta}{2} > \frac{\Delta}{2}$. Hence $\mathcal{H}_\Delta$ automatically holds in this case.

Otherwise, suppose that the conclusion of Lemma 19 holds for all $i \in [K]$ (which happens with probability $1 - K\delta$ conditioning on $\mathcal{G}_\Delta$ and $\mathcal{H}_{2\Delta}$). As we only pick those coordinates with $|\overline{r}_i| > \Delta$, all coordinates with magnitude at last $\Delta + \frac{\Delta}{4} + 3\Delta^2 < 3\Delta$ will be picked as $\Delta \leq \frac{1}{4}$, so the first

condition of $\mathcal{H}_\Delta$ indeed holds. Moreover, all picked coordinates will have magnitudes at least $\Delta - (\frac{\Delta}{4} + 3\Delta^2)$. But when $\Delta = \frac{1}{4}$, there will be no coordinates in $S$ as it is initially empty. And after that, we will have $\Delta^2 \le \frac{\Delta}{8}$. Hence, all coordinates with magnitude $\frac{\Delta}{2}$ will surely be identified, which means the second condition of $\mathcal{H}_\Delta$ also holds.

We then have $\Pr\{\mathcal{H}_\Delta \mid \mathcal{G}_\Delta, \mathcal{H}_{2\Delta}\} \ge 1 - K\delta$ by putting these two cases together. $\qquad\square$

### F.4 SINGLE-PHASE REGRET

*Proof of Lemma 21.* Conditioning on $\mathcal{G}_\Delta$, as we are playing $x_{t,i} = \widehat{\theta}_i$ for all $i \in S$ and $t \in \mathcal{T}_\Delta^b$, we will have

$$\sum_{t \in \mathcal{T}_\Delta^b} \sum_{i \in S} (\theta_i^* - x_{t,i})\theta_i^* = \sum_{t \in \mathcal{T}_\Delta^b} \sum_{i \in S} (\theta_i^* - \widehat{\theta}_i)\theta_i^* < n_\Delta^b \cdot \Delta^2.$$

Now consider a single coordinate not in $S$. For each $t \in \mathcal{T}_\Delta^b$, we will equiprobably play $\pm\frac{R}{\sqrt{K}}$ for this coordinate. Hence, the total regret will become

$$\sum_{t \in \mathcal{T}_\Delta^b} \left(\theta_i^* \pm \frac{R}{\sqrt{K}}\right)\theta_i^* = n_\Delta^b \cdot (\theta_i^*)^2 + \theta_i^* \sum_{t \in \mathcal{T}_\Delta^b} \left(\pm\frac{R}{\sqrt{K}}\right).$$

By $\mathcal{H}_{2\Delta}$, the first term is bounded by $36n_\Delta^b\Delta^2$. By Chernoff bound, the absolute value of the summation in the second term will be bounded by $\frac{R}{\sqrt{K}}\sqrt{2n_\Delta^b \ln\frac{1}{\delta}}$ with probability $1 - \delta$. As there are at most $s$ non-zero coordinates by sparsity, from a union bound, we can conclude that

$$\sum_{t \in \mathcal{T}_\Delta^b} \langle \theta^* - x_t, \theta^* \rangle \le 36sn_\Delta^b\Delta^2 + 6s\Delta\frac{R}{\sqrt{K}}\sqrt{2n_\Delta^b \ln\frac{1}{\delta}}$$

with probability $1 - s\delta$. $\qquad\square$

## G OMITTED PROOF IN SECTION 4.1 (ANALYSIS OF VOFUL2)

### G.1 PROOF OF MAIN THEOREM

In this section, we prove Theorem 5, which is restated as Theorem 22. We first assume that Proposition 4 is indeed correct, whose discussion is left to Appendix G.2.

**Theorem 22** (Regret of Algorithm 1 with VOFUL2 in Unkown-Variance Case). *Consider Algorithm 1 with $\mathcal{F}$ as VOFUL2 (Kim et al., 2021) and $\overline{\mathcal{R}_n^\mathcal{F}}$ as*

$$\overline{\mathcal{R}_n^\mathcal{F}} = C\left(s^{1.5}\sqrt{\sum_{k=1}^n (r_k - \langle x_k, \widehat{\beta}\rangle)^2 \ln\frac{1}{\delta}} + s^2\sqrt{2\ln\frac{n}{s\delta^2}\ln\frac{1}{\delta}} + s^{1.5}\sqrt{2\ln\frac{1}{\delta}} + s^2\ln\frac{1}{\delta}\right), \quad (20)$$

*where $C = \widetilde{\mathcal{O}}(1)$ is the constant hidden in the $\widetilde{\mathcal{O}}$ notation of Proposition 4, $x_1, x_2, \ldots, x_n$ are the actions made by the agent, $r_1, r_2, \ldots, r_n$ are the corresponding (noisy) feedback, and $\widehat{\beta}$ is defined as*

$$\widehat{\beta} = \operatorname*{argmin}_{\beta \in \mathbb{R}^s} \left(\sum_{k=1}^n (r_k - \langle x_k, \beta\rangle)^2 + \|\beta\|_2\right).$$

*The algorithm ensures the following regret bound with probability $1 - \delta$:*

$$\mathcal{R}_T = \widetilde{\mathcal{O}}\left((s^{2.5} + s\sqrt{d})\sqrt{\sum_{t=1}^T \sigma_t^2 \log\frac{1}{\delta}} + s^3\log\frac{1}{\delta}\right).$$

*Proof.* By Proposition 4, the total regret incurred by algorithm $\mathcal{F}$ satisfies

$$\mathcal{R}_n^{\mathcal{F}} \leq C \left( s^{1.5} \sqrt{\sum_{k=1}^{n} \eta_k^2 \ln \frac{1}{\delta}} + s^2 \ln \frac{1}{\delta} \right),$$

where $C = \widetilde{\mathcal{O}}(1)$ is a constant (with some logarithmic factors) and $\eta_k \sim \mathcal{N}(0, \sigma_k^2)$ is the noise for the $k$-th round executing $\mathcal{F}$. We now consider our regret estimator $\overline{\mathcal{R}_n^{\mathcal{F}}}$. We show that, with high probability, it is a pessimistic estimation.

**Lemma 23.** *For any given $\Delta$, with probability $1 - \delta$, $\overline{\mathcal{R}_n^{\mathcal{F}}} \geq \mathcal{R}_n^{\mathcal{F}}$.*

Therefore, for the "explore" phase, we can make use of Theorem 16 which only requires $\overline{\mathcal{R}_n^{\mathcal{F}}}$ to be an over-estimate of $\mathcal{R}_n^{\mathcal{F}}$ w.h.p., giving

$$\mathcal{R}_T^b \leq \widetilde{\mathcal{O}} \left( s\sqrt{d} \sqrt{\sum_{t=1}^{T} \sigma_t^2 \log \frac{1}{\delta}} + s \log \frac{1}{\delta} \right).$$

So we only need to bound the regret for the "commit" phase, namely $\mathcal{R}_T^a$. As mentioned in the main text, we will consider the regret contributed from inside and outside $S$ seprately. Formally, we will write $\mathcal{R}_T^a$ as

$$\mathcal{R}_T^a = \sum_{\Delta=2^{-2},\ldots,\Delta_f} \sum_{t \in \mathcal{T}_\Delta^a} \left( \sum_{i \in S} \theta_i^* (\theta_i^* - x_{t,i}) + \sum_{i \notin S} (\theta_i^*)^2 \right),$$

where the equality is because we will not put any mass on those $i \notin S$ during the "commit" phase. We still assume the 'good events' $\mathcal{G}_\Delta, \mathcal{H}_\Delta$ hold for all $\Delta$ (defined in Eq. (10)). By $\mathcal{H}_{2\Delta}$, for a given $\Delta$, $\sum_{i \notin S} (\theta_i^*)^2 \leq 36s\Delta^2$. Moreover, by Lemma 23, we will have

$$\mathcal{R}_T^a \leq \sum_{\Delta=2^{-2},\ldots,\Delta_f} \overline{\mathcal{R}_{n_\Delta^a}^{\mathcal{F}}} + 36s \sum_{\Delta=2^{-2},\ldots,\Delta_f} n_\Delta^a \Delta^2.$$

By the terminating criterion $\frac{1}{n} \overline{\mathcal{R}_n^{\mathcal{F}}} < \Delta^2$ (Line 5 of Algorithm 1), for all $\Delta$, we will have $n_\Delta^a - 1 \leq \frac{C}{\Delta^2} \overline{\mathcal{R}_{n_\Delta^a}^{\mathcal{F}}}$, which means

$$n_\Delta^a \leq \frac{C}{\Delta^2} \left( s^{1.5} \sqrt{\sum_{t \in \widetilde{\mathcal{T}}_\Delta^a} (r_t - \langle x_t, \widehat{\beta} \rangle)^2 \ln \frac{1}{\delta}} + s^2 \sqrt{2 \ln \frac{n_\Delta^a}{s\delta^2} \ln \frac{1}{\delta}} + s^{1.5} \sqrt{2 \ln \frac{1}{\delta}} + s^2 \ln \frac{1}{\delta} \right) + 1.$$

Therefore, plugging back into the expression of $\mathcal{R}_T^a$ gives (the term related with $\overline{\mathcal{R}_{n_\Delta^a}^{\mathcal{F}}}$ is dominated by the second term, as intuitively explained in the main text):

$$\mathcal{R}_T^a \leq \sum_{\Delta=2^{-2},\ldots,\Delta_f} \widetilde{\mathcal{O}} \left( s^{2.5} \sqrt{\sum_{t \in \widetilde{\mathcal{T}}_\Delta^a} (r_t - \langle x_t, \widehat{\beta} \rangle)^2 \ln \frac{1}{\delta}} + s^3 \left( \sqrt{\ln \frac{1}{\delta}} + \ln \frac{1}{\delta} \right) + s\Delta^2 \right).$$

The last term is simply bounded by $\mathcal{O}(s)$ after summing up over $\Delta$. Let us focus on the first term, where we need to upper bound the magnitude of $\overline{\mathcal{R}_n^{\mathcal{F}}}$. Applying Lemma 15 shows that the following with probability $1 - \delta$:

$$\sum_{k=1}^{n} \left( r_k - \langle x_k, \widehat{\beta} \rangle \right)^2 \leq \sum_{k=1}^{n} \eta_k^2 + 2s \ln \frac{n}{s\delta^2} + 2.$$

Therefore, we can write the regret (conditioning on this good event holds for all $\Delta$, which is with probability at least $1 - dT\delta$ according to Theorem 16) as

$$\mathcal{R}_T^a \leq \sum_{\Delta=2^{-2},\ldots,\Delta_f} \widetilde{\mathcal{O}}\left( s^{2.5}\sqrt{\sum_{t\in\mathcal{T}_\Delta^a} \eta_t^2 \log\frac{1}{\delta}} + s^3\left(\sqrt{\log\frac{1}{\delta}} + \log\frac{1}{\delta}\right) + s\Delta^2 \right). \quad (21)$$

Again by the technique we used for Appendix F, we will need the following lower bound for all $n_\Delta^a$ except for the last one, which follows from the fact that $\overline{\mathcal{R}_n^\mathcal{F}} \geq \mathcal{R}_n^\mathcal{F}$ (Lemma 23):

$$n_\Delta^a \geq \frac{Cs^{1.5}}{\Delta^2}\sqrt{\sum_{t\in\widetilde{\mathcal{T}}_\Delta^a} \eta_t^2 \ln\frac{1}{\delta}}.$$

By the summation technique that we used in Appendix F, more preciously, by the derivation of Eq. (15), we will have the following derivation:

$$\sum_{\Delta\neq\Delta_f} Cs^{1.5}\sqrt{\sum_{t\in\mathcal{T}_\Delta^a} \eta_t^2 \ln\frac{1}{\delta}} \leq \sqrt{\log_4 X}\sqrt{\sum_{\frac{1}{4}\geq\Delta\geq\Delta_X}\sum_{t\in\mathcal{T}_\Delta^a} C^2 s^3 \eta_t^2 \ln\frac{1}{\delta}} + \frac{1}{X}\sum_{\Delta_X>\Delta>\Delta_f} \frac{Cs^{1.5}}{\Delta^2}\sqrt{\sum_{t\in\mathcal{T}_\Delta^a} \eta_t^2 \ln\frac{1}{\delta}}$$

where $X$ is defined as

$$X = T\Big/\sqrt{\sum_{\Delta\neq\Delta_f}\sum_{t\in\mathcal{T}_\Delta^a} C^2 s^3 \eta_t^2 \ln\frac{1}{\delta}}$$

and $\Delta_X = 2^{-\lceil\log_4 X\rceil}$, which means $\Delta_X^2 \leq \frac{1}{X}$. Hence, we will have (the second summation will be bounded by $\frac{T}{X}$ as $\sum_\Delta n_\Delta^a \leq T$)

$$\sum_{\Delta\neq\Delta_f} Cs^{1.5}\sqrt{\sum_{t\in\mathcal{T}_\Delta^a} \sigma_t^2 \ln\frac{1}{\delta}} \leq \widetilde{\mathcal{O}}\left(\sqrt{\sum_{\Delta\neq\Delta_f}\sum_{t\in\mathcal{T}_\Delta^a} C^2 s^3 \eta_t^2 \log\frac{1}{\delta}}\right) = \widetilde{\mathcal{O}}\left(s^{1.5}\sqrt{\sum_{t=1}^T \eta_t^2 \log\frac{1}{\delta}}\right).$$

In other words, the first term of Eq. (21) will be bounded by

$$s^{2.5}\sum_{\Delta=2^{-2},\ldots,\Delta_f}\sqrt{\sum_{t\in\mathcal{T}_\Delta^a} \eta_t^2 \ln\frac{1}{\delta}} \leq \widetilde{\mathcal{O}}\left(s^{2.5}\sqrt{\sum_{t=1}^T \eta_t^2 \log\frac{1}{\delta}}\right).$$

So we are done with the first and the last term of Eq. (21). Now consider the second term, which is equivalent to bounding $\sum_\Delta 1$. Use the following property guaranteed by (again) Line 5 of Algorithm 1:

$$\sum_{\Delta\neq\Delta_f} \Delta^{-2} C s^2 \ln\frac{4}{\delta} \leq \sum_{\Delta\neq\Delta_f} n_\Delta^a \leq T,$$

which means $\sum_\Delta 1 \leq \log_4(T/s^2) = \widetilde{\mathcal{O}}(1)$. Combining them together gives

$$\mathcal{R}_T^a \leq \widetilde{\mathcal{O}}\left(s^{2.5}\sqrt{\sum_{t=1}^T \eta_t^2 \log\frac{1}{\delta}} + s^3\left(\sqrt{\log\frac{1}{\delta}} + \log\frac{1}{\delta}\right)\right),$$

At last, due to Theorem 13, with probability $1 - \delta$,

$$\sum_{i=1}^n (r_i - \langle x_i, \beta^*\rangle)^2 = \sum_{k=1}^n \eta_k^2 \overset{(a)}{\leq} \sum_{k=1}^n \mathbb{E}[\eta_k^2 \mid \mathcal{F}_{k-1}] + 4\sqrt{2}\sum_{k=1}^n \sigma_k^2 \ln\frac{2}{\delta} \leq 8\sum_{k=1}^n \sigma_k^2 \ln\frac{2}{\delta},$$

---

**Algorithm 3** VOFUL2 Algorithm (Kim et al., 2021)

---
1: **for** $t = 1, 2, \ldots, T$ **do**
2:     Compute the action for the $t$-th round as

$$x_t = \underset{x \in \mathcal{X}}{\operatorname{argmax}} \max_{\theta \in \bigcap_{s=1}^{t-1} \Theta_s} \langle x, \theta \rangle, \tag{22}$$

    where $\Theta_s$ is defined in Eq. (24).
3:     Observe reward $r_t = \langle x_t, \theta^* \rangle + \eta_t$.

---

where (a) is due to Theorem 13 and (b) is due to $\mathbb{E}[\eta_k^2 \mid \mathcal{F}_{k-1}] \leq \sigma_k^2$. Hence,

$$\mathcal{R}_T^a \leq \widetilde{\mathcal{O}} \left( s^{2.5} \sqrt{\sum_{t=1}^T \sigma_t^2 \log \frac{1}{\delta}} + s^3 \left( \sqrt{\log \frac{1}{\delta}} + \log \frac{1}{\delta} \right) \right).$$

Combining this with the regret for $\mathcal{R}_T^b$ (Theorem 16) gives

$$\mathcal{R}_T \leq \mathcal{O} \left( (s^{2.5} + s\sqrt{d}) \sqrt{\sum_{t=1}^T \sigma_t^t \log \frac{1}{\delta}} + s^3 \left( \sqrt{\log \frac{1}{\delta}} + \log \frac{1}{\delta} \right) + s \log \frac{1}{\delta} \right)$$

$$= \mathcal{O} \left( (s^{2.5} + s\sqrt{d}) \sqrt{\sum_{t=1}^T \sigma_t^t \log \frac{1}{\delta}} + s^3 \log \frac{1}{\delta} \right),$$

as claimed. $\qquad\qquad\qquad\qquad\qquad\qquad\qquad\qquad\qquad\qquad\qquad\qquad\qquad\qquad\qquad\qquad\quad\square$

## G.2 REGRET OVER-ESTIMATION

In this section, we first discuss why the strengthened Proposition 4 holds. After that, we argue that our pessimistic estimation $\overline{\mathcal{R}_n^{\mathcal{F}}}$ is indeed an over-estimation of $\mathcal{R}_n^{\mathcal{F}}$. We state the VOFUL2 algorithm in Algorithm 3 and also restate Proposition 4 as Proposition 24 for the ease of presentation.

**Proposition 24.** *VOFUL2 on $d$ dimensions guarantees, with probability at least $1 - \delta$,*

$$\mathcal{R}_T^{\mathcal{F}} = \widetilde{\mathcal{O}} \left( d^{1.5} \sqrt{\sum_{t=1}^T \eta_t^2 \log \frac{1}{\delta}} + d^2 \log \frac{1}{\delta} \right) = \widetilde{\mathcal{O}} \left( d^{1.5} \sqrt{\sum_{t=1}^T \sigma_t^2 \log \frac{1}{\delta}} + d^2 \log \frac{1}{\delta} \right), \tag{23}$$

*where $T$ is a stopping time finite a.s. and $\sigma_1^2, \sigma_2^2, \ldots, \sigma_T^2$ are the variances of $\eta_1, \eta_2, \ldots, \eta_T$.*

*Proof.* We will follow the proof of Kim et al. (2021) and highlight the different steps. We first argue that an analog to their Empirical Bernstein Inequality (Zhang et al., 2021, Theorem 4) still holds.

**Lemma 25** (Analog of Theorem 4 from Zhang et al. (2021))**.** *Let $\{X_i\}_{i=1}^n$ be a sequence of zero-mean Gaussian random variables such that $n$ is a stopping time finite a.s. Then for all $n \geq 8$ and any $\delta \in (0, 1)$,*

$$\Pr \left\{ \left| \sum_{i=1}^n X_i \right| \leq 8 \sqrt{\sum_{i=1}^n X_i^2 \ln \frac{4}{\delta}} \right\} \geq 1 - \delta.$$

*Proof.* This is a direct corollary of Theorem 10. $\qquad\qquad\qquad\qquad\qquad\qquad\qquad\qquad\qquad\quad\square$

Thanks to Theorem 14, the following analog of Lemma 17 from Zhang et al. (2021) also holds:

**Lemma 26** (Analog of Lemma 17 from Zhang et al. (2021)). *Let $\{X_i\}_{i=1}^n$ be a sequence of zero-mean Gaussian random variables such that $n$ is a stopping time finite a.s. Then for all $\delta \in (0, 1)$,*

$$\Pr\left\{\sum_{i=1}^n X_i^2 \geq 8 \sum_{i=1}^n \sigma_i^2 \ln \frac{2}{\delta}\right\} \leq \delta,$$

*where $\sigma_i^2$ is the variance of $X_i$.*

Then consider their Elliptical Potential Counting Lemma (Kim et al., 2021, Lemma 5), which holds as long as $\|x_t\|_2 \leq 1$. In our setting, we indeed have this property as $x_t \in \mathbb{B}^d$ (only the noises can be unbounded, instead of actions). Hence, this lemma still holds.

**Proposition 27** (Kim et al. (2021, Lemma 5)). *Let $x_1, x_2, \ldots, x_k \in \mathbb{R}^d$ be such that $\|x_i\|_2 \leq 1$ for all $s \in [k]$. Let $V_k = \lambda I + \sum_{i=1}^k x_i x_i^\mathsf{T}$. Let $J = \{i \in [k] \mid \|x_i\|_{V_{i-1}^{-1}}^2 \geq q\}$, then*

$$|J| \leq \frac{2d}{\ln(1+q)} \ln\left(1 + \frac{2/e}{\ln(1+q)} \frac{1}{\lambda}\right).$$

Then consider the confidence set construction:

$$\Theta_t = \bigcap_{\ell=1}^L \left\{ \theta \in \mathbb{B}^d \left| \left| \sum_{s=1}^t \overline{(x_s^\mathsf{T}\mu)}_\ell \epsilon_s(\theta) \right| \leq \sqrt{\sum_{s=1}^t \overline{(x_s^\mathsf{T}\mu)}_\ell^2 \epsilon_s^2(\theta)\iota}, \forall \mu \in \mathbb{B}^d \right. \right\}, \tag{24}$$

where $\epsilon_s(\theta) = r_s - x_s^\mathsf{T}\theta$, $\epsilon_s^2(\theta) = (\epsilon_s(\theta))^2$, $\iota = 128\ln((12K2^L)^{d+2}/\delta) = \mathcal{O}(\sqrt{d})$, $L = \max\{1, \lfloor\log_2(1 + \frac{T}{d})\rfloor\}$ and $\overline{(x)}_\ell = \min\{|x|, 2^{-\ell}\}\frac{x}{|x|}$. By using Lemma 25 together with their original $\epsilon$-net coverage argument, we have $\theta^* \in \Theta_t$ for all $t \in [T]$ with high probability:

**Lemma 28** (Analog of Lemma 1 from Kim et al. (2021)). *The good event $\mathcal{E}_1 = \{\forall t \in [T], \theta^* \in \Theta_t\}$ happens with probability $1 - \delta$.*

Similar to Kim et al. (2021), we define $\theta_t$ be the maximizer of Eq. (22) in the $t$-th round and define $\mu_t = \theta_t - \theta^*$. We also consider the following good event

$$\mathcal{E}_2 : \forall t \in [T], \sum_{s=1}^t \epsilon_s^2(\theta^*) \leq 8 \sum_{s=1}^t \sigma_s^2 \ln \frac{8T}{\delta},$$

which happens with probability $1 - \delta$ due to Lemma 26. Define

$$W_{\ell,t-1}(\mu) = 2^{-\ell}\lambda I + \sum_{s=1}^{t-1} \left(1 \wedge \frac{2^{-\ell}}{|x_s^\mathsf{T}\mu|}\right) x_s x_s^\mathsf{T}.$$

Abbreviate $W_{\ell,t-1}(\mu_t)$ as $W_{\ell,t-1}$, then we have the following lemma, which is slightly different from the original one, whose proof will be presented later.

**Lemma 29** (Analog of Lemma 4 from Kim et al. (2021)). *Conditioning on $\mathcal{E}_1$ and setting $\lambda = 1$, we have*

1. *$\|\mu_t\|_{W_{\ell,t-1}}^2 \leq C_1 2^{-\ell}(\sqrt{A_{t-1}\iota} + \iota)$ for some absolute constant $C_1$, where $A_t \triangleq \sum_{s=1}^t \eta_s^2$.*
2. *For all $s \leq t$, we have $\|\mu_t\|_{W_{\ell,s-1}}^2 \leq C_1 2^{-\ell}(\sqrt{A_{s-1}\iota} + \iota)$.*
3. *There exists absolute constant $C_2$ such that $x_t\mu_t \leq C_2\|x_t^\mathsf{T}\|_{W_{\ell,t-1}^{-1}}^2(\sqrt{A_{t-1}\iota} + \iota)$.*

Therefore, as we have the same $\mathcal{E}_1$, Lemma 5 and a similar Lemma 4 (which uses $\eta_s$ instead of $\sigma_s$), we can conclude that

$$\mathcal{R}_T^\mathcal{F} \leq C\left(\sqrt{\sum_{s=1}^{T-1} \eta_s^2 \iota} + \iota\right) d\ln^2\left(1 + C\left(\sqrt{\sum_{s=1}^{T-1} \eta_s^2 \iota} + \iota\right)\left(1 + \frac{T}{d}\right)^2\right)$$

$$= \widetilde{\mathcal{O}} \left( d^{1.5} \sqrt{\sum_{t=1}^{T} \eta_t^2 \log \frac{1}{\delta}} + d^2 \log \frac{1}{\delta} \right),$$

as in Kim et al. (2021, Theorem 2) (recall that $\iota = \mathcal{O}(\sqrt{d})$). Conditioning on $\mathcal{E}_2$, we then further have

$$\mathcal{R}_T^{\mathcal{F}} \leq \widetilde{\mathcal{O}} \left( d^{1.5} \sqrt{\sum_{t=1}^{T} \sigma_t^2 \log \frac{1}{\delta}} + d^2 \log \frac{1}{\delta} \right),$$

as claimed. □

*Proof of Lemma 29.* By definition and abbreviating $\overline{(\cdot)}_\ell$ as $\overline{(\cdot)}$, we have

$$\|\mu_t\|_{W_{\ell,t-1}-2^{-\ell}\lambda I}^2 = \sum_{s=1}^{t-1} \overline{(x_s^\mathsf{T} \mu_t)}(x_s^\mathsf{T} \mu_t) = \sum_{s=1}^{t-1} \overline{(x_s^\mathsf{T} \mu_s)}(x_s \theta_t - r_t + r_t - x_s \theta^*)$$

$$= \sum_{s=1}^{t-1} \overline{(x_s^\mathsf{T} \mu_t)}(-\epsilon_s(\theta_t) + \epsilon_s(\theta^*))$$

$$\overset{(\mathcal{E}_1)}{\leq} \sqrt{\sum_{s=1}^{t-1} (x_s^\mathsf{T} \mu_t)^2 \epsilon_s^2(\theta_t) \iota} + \sqrt{\sum_{s=1}^{t-1} \overline{(x_s^\mathsf{T} \mu_t)}^2 \epsilon_s^2(\theta^*) \iota}$$

$$\overset{(a)}{\leq} \sqrt{\sum_{s=1}^{t-1} \overline{(x_s^\mathsf{T} \mu_t)}^2 2(x_s^\mathsf{T} \mu_t)^2 \iota} + 2\sqrt{\sum_{s=1}^{t-1} \overline{(x_s^\mathsf{T} \mu_t)}^2 2\epsilon_s^2(\theta^*) \iota}$$

$$\overset{(b)}{\leq} \sqrt{2^{-\ell} \sum_{s=1}^{t-1} \overline{(x_s^\mathsf{T} \mu_t)}^2 2(x_s^\mathsf{T} \mu_t)^2 \iota} + 2^{-\ell} 16 \sqrt{\sum_{s=1}^{t-1} \eta_s^2 \iota}$$

$$\leq 2\sqrt{2 \sum_{s=1}^{t-1} \overline{(x_s^\mathsf{T} \mu_t)}(x_s^\mathsf{T} \mu_t) \iota} + 2^{-\ell} 16 \sqrt{\sum_{s=1}^{t-1} \eta_s^2 \iota}$$

$$= \sqrt{2^{-\ell} 8 \|\mu_t\|_{V_{t-1}-\lambda I}^2 \iota} + 2^{-\ell} 16 \sqrt{\sum_{s=1}^{t-1} \eta_s^2 \iota},$$

where (a) used $\epsilon_s^2(\theta_t) = (r_s - x_s \theta_t)^2 = (x_s^\mathsf{T}(\theta^* - \theta_t) + \epsilon_s^2(\theta^*)) \leq 2(x_s^\mathsf{T} \mu_t)^2 + 2\epsilon_s^2$ and (b) used $\epsilon_s^2(\theta^*) = \eta_s^2$. By the self-bounding property Lemma 37, we have

$$\|\mu_t\|_{V_{t-1}-\lambda I}^2 \leq 16 \sqrt{\sum_{s=1}^{t-1} \sigma_s^2} + 8\iota,$$

which means $\|\mu_t\|_{V_{t-1}}^2 \leq 4\lambda + 16 \sqrt{\sum_{s=1}^{t-1} \sigma_s^2 \iota} + 8\iota$. Setting $\lambda = 1$ gives the first conclusion. Based on this, the second and third conclusion directly follow according to Kim et al. (2021). □

## G.3 EXTENSION TO UNKNOWN VARIANCE CASES

Based on Proposition 4, we then show that, our regret estimation $\overline{\mathcal{R}_n^{\mathcal{F}}}$ Eq. (20) is indeed pessimistic (i.e., Lemma 23).

*Proof of Lemma 23.* From Lemma 15 with $\lambda = 1$, with probability $1 - \delta$, we will have (recall the assumption that $\sigma_t^2 \leq 1$ for all $t \in [T]$)

$$\sum_{k=1}^{n} (r_k - \langle x_k, \beta^* \rangle)^2 = \sum_{k=1}^{n} \eta_k^2 \leq \sum_{k=1}^{n} (r_k - \langle x_k, \widehat{\beta} \rangle)^2 + 2s^2 \ln \frac{n}{s\delta^2} + 2.$$

Therefore we have

$$
\begin{aligned}
\mathcal{R}_n^{\mathcal{F}} &\leq C \left( s^{1.5} \sqrt{\sum_{k=1}^{n} \eta_k^2 \ln \frac{1}{\delta} + s^2 \ln \frac{1}{\delta}} \right) \\
&\leq C \left( s^{1.5} \sqrt{\left( \sum_{k=1}^{n} (r_k - \langle x_k, \widehat{\beta} \rangle)^2 + 2s \ln \frac{n}{s\delta^2} + 2 \right) \ln \frac{1}{\delta} + s^2 \ln \frac{1}{\delta}} \right) \\
&\leq C \left( s^{1.5} \sqrt{\sum_{k=1}^{n} (r_k - \langle x_k, \widehat{\beta} \rangle)^2 \ln \frac{1}{\delta}} + s^2 \sqrt{2 \ln \frac{n}{s\delta^2} \ln \frac{1}{\delta}} + s^{1.5} \sqrt{2 \ln \frac{1}{\delta}} + s^2 \ln \frac{1}{\delta} \right) = \overline{\mathcal{R}_n^{\mathcal{F}}}.
\end{aligned}
$$

In other words, our $\overline{\mathcal{R}_n^{\mathcal{F}}}$ is an over-estimation of $\mathcal{R}_n^{\mathcal{F}}$ with probability $1 - \delta$. $\qquad\square$

# H OMITTED PROOF IN SECTION 4.2 (ANALYSIS OF WEIGHTED OFUL)

## H.1 PROOF OF MAIN THEOREM

Similar to the VOFUL2 algorithm, we still assume Proposition 6 indeed holds and defer the discussions to the next section. We restate the regret guarantee of Weighted OFUL (Zhou et al., 2021), namely Theorem 7, for the ease of reading, as follows:

**Theorem 30** (Regret of Algorithm 1 with Weighted OFUL in Known-Variance Case). *Consider Algorithm 1 with $\mathcal{F}$ as* Weighted OFUL *(Zhou et al., 2021) and $\overline{\mathcal{R}_n^{\mathcal{F}}}$ as*

$$
\overline{\mathcal{R}_n^{\mathcal{F}}} \triangleq C \left( \sqrt{sn \ln \frac{1}{\delta}} + s \sqrt{\sum_{k=1}^{n} \sigma_k^2 \ln \frac{1}{\delta}} \right).
$$

*The algorithm ensures the following regret bound with probability $1 - \delta$:*

$$
\mathcal{R}_T = \widetilde{\mathcal{O}} \left( (s^2 + s\sqrt{d}) \sqrt{\sum_{t=1}^{T} \sigma_t^2 \log \frac{1}{\delta}} + s^{1.5} \sqrt{T} \log \frac{1}{\delta} \right).
$$

*Proof.* Firstly, from Proposition 6, the condition of applying Theorem 16 holds. Therefore, we have

$$
\mathcal{R}_T^b \leq \widetilde{\mathcal{O}} \left( s\sqrt{d} \sqrt{\sum_{t=1}^{T} \sigma_t^2 \log \frac{1}{\delta}} + s \log \frac{1}{\delta} \right).
$$

For $\mathcal{R}_T^a$, similar to Appendix H, we decompose it into two parts: those from $S$ and from outside of $S$. The former case is bounded by Proposition 6, as

$$
\text{Regret from } S \text{ with gap threshold } \Delta \leq C \left( \sqrt{sn_\Delta^a \ln \frac{1}{\delta}} + s \sqrt{\sum_{t \in \mathcal{T}_\Delta^a} \sigma_t^2 \ln \frac{1}{\delta}} \right).
$$

For those outside $S$, we will bound it as $\mathcal{O}(sn_\Delta^a \Delta^2)$, where we only need to bound $n_\Delta^a$. From Line 5 of Algorithm 1, we have

$$
n_\Delta^a - 1 \leq \frac{C}{\Delta^2} \left( \sqrt{s(n_\Delta^a - 1) \ln \frac{1}{\delta}} + s \sqrt{\sum_{t \in \widetilde{\mathcal{T}_\Delta^a}} \sigma_t^2 \ln \frac{1}{\delta}} \right).
$$

By the "self-bounding" property that $x \le a + b\sqrt{x}$ implies $x \le \mathcal{O}(a + b^2)$ (Lemma 37), we have

$$
n_\Delta^a - 1 \le \frac{1}{\Delta^4} s \ln \frac{1}{\delta} + \frac{s}{\Delta^2} \sqrt{\sum_{t \in \widetilde{\mathcal{T}}_\Delta^a} \sigma_t^2 \ln \frac{1}{\delta}}.
$$

Therefore, we can conclude that (the regret from $S$ is dominated)

$$
\mathcal{R}_T^a \le \sum_{\Delta = 2^{-2}, \dots} \mathcal{O}\left( \frac{s^2}{\Delta^2} \log \frac{1}{\delta} + s^2 \sqrt{\sum_{t \in \widetilde{\mathcal{T}}_\Delta^a} \sigma_t^2 \log \frac{1}{\delta}} + s\Delta^2 \right),
$$

where the last term is simply bounded by $\mathcal{O}(s)$. Again from Line 5 of Algorithm 1, we will have the following property for all $\Delta \ne \Delta_f$:

$$
n_\Delta^a > \frac{C}{\Delta^2} \left( \sqrt{s n_\Delta^a \ln \frac{1}{\delta}} + s \sqrt{\sum_{t \in \mathcal{T}_\Delta^a} \sigma_t^2 \ln \frac{1}{\delta}} \right). \tag{25}
$$

We first bound the second term, which basically follow the summation technique (Eq. (15)) that we used in Appendices F and G.1:

$$
\sum_{\Delta \ne \Delta_f} Cs \sqrt{\sum_{t \in \mathcal{T}_\Delta^a} \sigma_t^2 \ln \frac{1}{\delta}} \le \sqrt{\log_4 X} \sqrt{\sum_{\frac{1}{4} \ge \Delta \ge \Delta_X} \sum_{t \in \mathcal{T}_\Delta^a} C^2 s^2 \sigma_t^2 \ln \frac{1}{\delta}} + \frac{1}{X} \sum_{\Delta_X > \Delta > \Delta_f} \frac{Cs}{\Delta^2} \sqrt{\sum_{t \in \mathcal{T}_\Delta^a} \sigma_t^2 \ln \frac{1}{\delta}}
$$

where $X$ is defined as

$$
X = T \Big/ \sqrt{\sum_{\Delta \ne \Delta_f} \sum_{t \in \mathcal{T}_\Delta^a} C^2 s^2 \sigma_t^2 \ln \frac{1}{\delta}}
$$

and $\Delta_X = 2^{-\lceil \log_4 X \rceil}$, which means $\Delta_X^2 \le \frac{1}{X}$. Hence, we will have (the second summation will be bounded by $\frac{T}{X}$ as $\sum_\Delta n_\Delta^a \le T$)

$$
\sum_{\Delta \ne \Delta_f} Cs \sqrt{\sum_{t \in \mathcal{T}_\Delta^a} \sigma_t^2 \ln \frac{1}{\delta}} \le \widetilde{\mathcal{O}}\left( \sqrt{\sum_{\Delta \ne \Delta_f} \sum_{t \in \mathcal{T}_\Delta^a} C^2 s^2 \sigma_t^2 \log \frac{1}{\delta}} \right) = \widetilde{\mathcal{O}}\left( s \sqrt{\sum_{t=1}^T \sigma_t^2 \log \frac{1}{\delta}} \right).
$$

Hence, for the second term, we have

$$
\mathcal{O}\left( s^2 \sqrt{\sum_{t=1}^T \sigma_t^2 \log \frac{1}{\delta} \log T} \right) = \widetilde{\mathcal{O}}\left( s^2 \sqrt{\sum_{t=1}^T \sigma_t^2 \log \frac{1}{\delta}} \right).
$$

At last, we consider the first term. From the same lower bound of $n_\Delta^a$ (Eq. (25)), we will have

$$
n_\Delta^a > \frac{C}{\Delta^2} \sqrt{s n_\Delta^a \ln \frac{1}{\delta}} \implies n_\Delta^a > \frac{C^2}{\Delta^4} s \ln \frac{1}{\delta}.
$$

By the fact that $\sum_{\Delta \ne \Delta_f} n_\Delta^a \le T$, we will have

$$
T \ge C^2 s \ln \frac{1}{\delta} \sum_{\Delta \ne \Delta_f} \Delta^{-4} = \mathcal{O}(1) C^2 s \ln \frac{1}{\delta} (2\Delta_f)^{-4}.
$$

Henceforth,

$$
\mathcal{O}\left( s^2 \log \frac{1}{\delta} \cdot \sum_{\Delta = 2^{-2}, \dots, \Delta_f} \Delta^{-2} \right) = \mathcal{O}\left( s^2 \log \frac{1}{\delta} \Delta_f^{-2} \right) \le \mathcal{O}\left( s^2 \log \frac{1}{\delta} \sqrt{\frac{T}{s \ln \frac{1}{\delta}}} \right) = \mathcal{O}\left( s^{1.5} \sqrt{T \log \frac{1}{\delta}} \right).
$$

---

**Algorithm 4** `Weighted OFUL` Algorithm (Zhou et al., 2021)

---

1: Intialize $A_0 \leftarrow \lambda I$, $c_0 \leftarrow 0$, $\widehat{\theta}_0 \leftarrow A_0^{-1} c_0$, $\widehat{\beta}_0 = 0$ and $\Theta_0 \leftarrow \{\theta \mid \|\theta - \widehat{\theta}_0\|_{A_0} \leq \widehat{\beta}_0 + \sqrt{\lambda} B\}$.
2: **for** $t = 1, 2, \ldots, T$ **do**
3:     Compute the action for the $t$-th round as

$$x_t = \underset{x \in \mathcal{X}}{\operatorname{argmax}} \, \max_{\theta \in \Theta_{t-1}} \langle x, \theta \rangle. \tag{26}$$

4:     Observe reward $r_t = \langle x_t, \theta^* \rangle + \eta_t$ and variance information $\sigma_t^2$, set $\overline{\sigma}_t = \max\{1/\sqrt{d}, \sigma_t\}$, set confidence radius $\widehat{\beta}_t$ as

$$\widehat{\beta}_t = 8\sqrt{d \ln \left(1 + \frac{t}{d\lambda \overline{\sigma}_{\min,t}^2}\right) \ln \frac{4t^2}{\delta}}, \tag{27}$$

    where $\overline{\sigma}_{\min,t} \triangleq \min_{s=1}^t \overline{\sigma}_s$.
5:     Calculate $A_t \leftarrow A_{t-1} + x_t x_t^{\mathsf{T}}/\overline{\sigma}_t^2$, $c_t \leftarrow c_{t-1} + r_t x_t/\overline{\sigma}_t^2$, $\widehat{\theta}_t \leftarrow A_t^{-1} c_t$ and $\Theta_t \leftarrow \{\theta \mid \|\theta - \widehat{\theta}_t\|_{A_t} \leq \widehat{\beta}_t + \sqrt{\lambda} B\}$.

---

Combining all above together gives

$$\mathcal{R}_T = \mathcal{R}_T^a + \mathcal{R}_T^b \leq \mathcal{O}\left(s^{1.5}\sqrt{T \log \frac{1}{\delta}} + s^2 \sqrt{\sum_{t=1}^T \sigma_t^2 \log \frac{1}{\delta}} + s\right) + \widetilde{\mathcal{O}}\left(s\sqrt{d}\sqrt{\sum_{t=1}^T \sigma_t^2 \log \frac{1}{\delta}} + s \log \frac{1}{\delta}\right)$$

$$\leq \widetilde{\mathcal{O}}\left((s^2 + s\sqrt{d})\sqrt{\sum_{t=1}^T \sigma_t^2 \log \frac{1}{\delta}} + s^{1.5}\sqrt{T} \log \frac{1}{\delta}\right),$$

as claimed. □

## H.2 Regret Over-estimation

We again briefly argue that Proposition 6 holds under our noise model. We present their algorithm in Algorithm 4.

**Proposition 31.** *With probability at least $1 - \delta$, `Weighted OFUL` executed for $T$ steps on $d$ dimensions guarantees*

$$\mathcal{R}_T^{\mathcal{F}} \leq C\left(\sqrt{dT \log \frac{1}{\delta}} + d\sqrt{\sum_{t=1}^T \sigma_t^2 \log \frac{1}{\delta}}\right)$$

*where $C = \widetilde{\mathcal{O}}(1)$, $T$ is a stopping time finite a.s., and $\sigma_1^2, \sigma_2^2, \ldots, \sigma_T^2$ are the variances of $\eta_1, \eta_2, \ldots, \eta_T$.*

*Proof Sketch.* We mainly follow the original proof by Zhou et al. (2021) and highlight the differences. We first highlight their Bernstein Inequality for vector-valued martingales also holds under our assumptions, as:

**Lemma 32** (Analog of Theorem 4.1 from Zhou et al. (2021)). *Let $\{x_i\}_{i=1}^n$ be sequence of $d$-dimensional random vectoes such that $\|x_t\|_2 \leq L$. Let $\{\eta_i\}_{i=1}^n$ be a sequence of independent, symmetric and $\{\sigma_i^2\}_{i=1}^n$-sub-Gaussian random variables. Let $r_t = \langle \theta^*, x_t \rangle + \eta_t$ for all $t \in [n]$. Set $Z_t = \lambda I + \sum_{s=1}^t x_s x_s^{\mathsf{T}}$, $b_t = \sum_{s=1}^t r_s x_s$ and $\theta_t = Z_t^{-1} b_t$. Let $n$ be a stopping time finite a.s. Then, $\forall \delta \in (0, 1)$,*

$$\Pr\left\{\left\|\sum_{s=1}^t x_s \eta_s\right\|_{Z_t^{-1}} \leq \beta_t, \|\theta_t - \theta^*\|_{Z_t} \leq \beta_t + \sqrt{\lambda}\|\theta^*\|_2, \forall t \in [n]\right\} \geq 1 - \delta,$$

*where* $\beta_t = 8\sigma\sqrt{d\ln(1+\frac{tL^2}{d\lambda})\ln\frac{8t^2}{\delta}}$.

The proof, which will be presented later, mainly follows from the idea of their proof of Theorem 4.1, except that we are using Proposition 11. Check the proof below for more details about this.

With this theorem, we can consequently conclude that the confidence construction is indeed valid by applying Lemma 32 to the sequence $\{\eta_t/\overline{\sigma}_t\}_{t\in[T]}$, which gives

$$\|\widehat{\theta}_t - \theta^*\|_{A_t} \le \widehat{\beta}_t + \sqrt{\lambda}\|\theta^*\|_2 \le \widehat{\beta}_t + \sqrt{\lambda}, \quad \forall t \in [T], \quad \text{with probability } 1 - \delta,$$

where $\widehat{\beta}_t = 8\sqrt{d\ln(1+\frac{t}{d\lambda\overline{\sigma}_{\min,t}^2})\ln\frac{4t^2}{\delta}}$, as defined in Eq. (27). Therefore, we can conclude their (B.19) from exactly the same argument, namely

$$\mathcal{R}_T^{\mathcal{F}} \le 2\sum_{t=1}^{T}\min\left\{1, \overline{\sigma}_t(\widehat{\beta}_{t-1} + \sqrt{\lambda})\|x_t/\overline{\sigma}_t\|_{A_{t-1}^{-1}}\right\}.$$

Similar to their proof, define $\mathcal{I}_1 = \{t \in [T] \mid \|x_t/\overline{\sigma}_t\|_{A_{t-1}^{-1}} \ge 1\}$ and $\mathcal{I}_2 = [T]\setminus\mathcal{I}_1$, then we have (where $\overline{\sigma}_{\min}$ is the abbreviation of $\overline{\sigma}_{\min,T} = \min_{s=1}^{T}\overline{\sigma}_s$)

$$|\mathcal{I}_1| = \sum_{t\in\mathcal{I}_1}\min\{1, \|x_t/\overline{\sigma}_t\|_{A_{t-1}^{-1}}^2\} \le \sum_{t=1}^{T}\min\{1, \|x_t/\overline{\sigma}_t\|_{A_{t-1}^{-1}}^2\} \le 2d\ln\left(1 + \frac{T}{d\lambda\overline{\sigma}_{\min}^2}\right),$$

where the last step uses Proposition 34 and the fact that $\|x_t/\overline{\sigma}_t\|_2 \le \overline{\sigma}_{\min}^{-1}$. Therefore, we are having the same Eq. (B.21) as theirs, which gives

$$\mathcal{R}_T^{\mathcal{F}} \le 2\sqrt{2d\ln\left(1+\frac{T}{d\lambda\overline{\sigma}_{\min}^2}\right)}\sqrt{\sum_{t=1}^{T}(\widehat{\beta}_{t-1}+\sqrt{\lambda})^2\overline{\sigma}_t^2 + 4d\ln\left(1+\frac{T}{d\lambda\overline{\sigma}_{\min}^2}\right)}.$$

By the choice of $\overline{\sigma}_t = \max\{1/\sqrt{d}, \sigma_t\}$ and $\lambda = 1$, we have

$$\ln\left(1 + \frac{T}{d\lambda\overline{\sigma}_{\min}^2}\right) \le \ln\left(1 + \frac{T}{d}\right) = \widetilde{\mathcal{O}}(1),$$

and that

$$\widehat{\beta}_t + \sqrt{\lambda} = \mathcal{O}\left(\sqrt{d\log T\log\frac{T}{\delta}} + 1\right) = \widetilde{\mathcal{O}}\left(\sqrt{d\log\frac{1}{\delta}}\right),$$

which gives

$$\mathcal{R}_T^{\mathcal{F}} = \widetilde{\mathcal{O}}\left(d\sqrt{\sum_{t=1}^{T}\overline{\sigma}_t^2\log\frac{1}{\delta}}\right) = \widetilde{\mathcal{O}}\left(d\sqrt{\sum_{t=1}^{T}\left(\frac{1}{d}+\sigma_t^2\right)\log\frac{1}{\delta}}\right)$$

$$= \widetilde{\mathcal{O}}\left(\sqrt{dT\log\frac{1}{\delta}} + d\sqrt{\sum_{t=1}^{T}\sigma_t^2\log\frac{1}{\delta}}\right),$$

as claimed, while the second step uses $\overline{\sigma}_t^2 = \min\{\frac{1}{d}, \sigma_t^2\} \le \frac{1}{d} + \sigma_t^2$. $\qquad\square$

*Proof of Lemma 32.* Their original proof mainly use the following two auxiliary results: The first one is the well-known Freedman inequality (Freedman, 1975), which is originally for bounded martingale difference sequences, while the second one is Lemma 11 from Abbasi-Yadkori et al. (2011). For the former one, from its variant for sub-Gaussian random variables (Proposition 11), we have:

**Corollary 33.** *Suppose that $\{\xi_i\}_{i=1}^n$ is a sequence of zero-mean random variables where $\xi_i \sim$ subG$(\sigma_i^2)$ for some sequence $\{\sigma_i\}_{i=1}^n$. Let $n$ be a stopping time finite a.s. Then for all $x, v > 0$ and $\lambda > 0$,*

$$\Pr\left\{\exists 1 \le k \le n : \sum_{i=1}^k \xi_i \ge x \wedge \sum_{i=1}^k V_i \le v^2\right\} \le \exp\left(-\frac{x^2}{2v^2}\right).$$

*Moreover, for any $\delta \in (0, 1)$, with probability $1 - \delta$, we have*

$$\sum_{i=1}^n \xi_i \le 2\sqrt{\sum_{i=1}^n \sigma_i^2 \ln \frac{2}{\delta}}.$$

*Proof.* The first conclusion is done by applying Proposition 11 optimally with $f(\lambda) = \frac{1}{2}\lambda^2$, $V_i = \sigma_i^2$ and $\lambda = \frac{x}{v^2}$. The second conclusion is consequently proved by taking $v^2 = \sum_{i=1}^n \sigma_i^2$ and $x = 2v\sqrt{\ln \frac{2}{\delta}}$. □

For their second auxiliary lemma (Abbasi-Yadkori et al., 2011, Lemma 11), one can see that the original lemma indeed holds for sub-Gaussian random variables. Therefore, we still have the following lemma:

**Proposition 34** (Abbasi-Yadkori et al. (2011, Lemma 11))**.** *Let $\{x_t\}_{t=1}^T$ be a sequence in $\mathbb{R}^d$ and define $V_t = \lambda I + \sum_{s=1}^t x_s x_s^\mathsf{T}$ for some $\lambda > 0$. Then, if we have $\|x_t\|_2 \le L$ for all $t \in [T]$, then*

$$\sum_{t=1}^T \min\left\{1, \|x_t\|_{V_{t-1}^{-1}}^2\right\} \le 2\log\frac{\det(V_t)}{\det(\lambda I)} \le 2d\ln\frac{d\lambda + TL^2}{d\lambda}.$$

Recall the definition that $Z_t = \lambda I + \sum_{s=1}^t x_s x_s^\mathsf{T}$. Further define $d_t = \sum_{s=1}^t x_i \eta_i$, $w_t = \|x_t\|_{Z_{t-1}^{-1}}$ and let $\mathcal{E}_t$ be the event that $\|d_s\|_{Z_{s-1}^{-1}} \le \beta_s$ for all $s \le t$. They proved the following lemma, which still applies to our case:

**Lemma 35** (Analog of Lemma B.3 from Zhou et al. (2021))**.** *With probability $1 - \frac{\delta}{2}$, with the definitions of $x_t$ and $\eta_t$ in Lemma 32, the following inequality holds for all $t \ge 1$:*

$$\sum_{s=1}^t \frac{2\eta_s x_s^\mathsf{T} Z_{s-1}^{-1} d_{s-1}}{1 + w_s^2} \mathbb{1}[\mathcal{E}_{s-1}] \le \frac{3}{4}\beta_t^2.$$

*Proof.* We only need to verify whether we can apply our Freedman's inequality to $\ell_s \triangleq \frac{2\eta_s x_s^\mathsf{T} Z_{s-1}^{-1} d_{s-1}}{1 + w_s^2} \mathbb{1}[\mathcal{E}_{s-1}]$. It is obvious that $\mathbb{E}[\ell_s \mid \mathcal{F}_{s-1}] = 0$. Moreover, from the following inequality (which is their Eq. (B.3))

$$|\ell_s| \le \frac{2\|x_s\|_{Z_{s-1}^{-1}}}{1 + w_s^2}\|d_{s-1}\|_{Z_{s-1}^{-1}}\mathbb{1}[\mathcal{E}_{s-1}] \le \frac{2w_i}{1 + w_i^2}\beta_{s-1} \le \min\{1, 2w_i\}\beta_{i-1},$$

and the fact that $\eta_s \mid \mathcal{F}_{s-1} \sim$ subG$(\sigma^2)$, we have $\ell_s \mid \mathcal{F}_{s-1} \sim$ subG$((\sigma\beta_{s-1}\min\{1, 2w_s\})^2)$. Denote the sub-Gaussian parameter as $\widetilde{\sigma}_s$ for simplicity. We have

$$\sum_{s=1}^t \widetilde{\sigma}_s^2 \le \sigma^2\beta_t^2 \sum_{s=1}^t (\min\{1, 2w_s\})^2 \le 4\sigma^2\beta_t^2 \sum_{s=1}^t \min\{1, w_s^2\} \le 8\sigma^2\beta_t^2 d\ln\left(1 + \frac{tL^2}{d\lambda}\right),$$

where the first inequality is due to the non-decreasing property of $\{\beta_s\}$ and the last one is due to Proposition 34. Therefore, from our Freedman's inequality (Corollary 33), we can conclude that with probability $1 - \delta/(4t^2)$,

$$\sum_{s=1}^t \ell_s \le 2\sqrt{\sum_{s=1}^t 8\sigma^2\beta_t^2 d\ln\left(1 + \frac{tL^2}{d\lambda}\right)\ln\frac{8t^2}{\delta}}$$

$$\leq \frac{\beta_t^2}{2} + 16\sigma^2 d \ln\left(1 + \frac{tL^2}{d\lambda}\right) \ln \frac{8t^2}{\delta} = \frac{3}{4}\beta_t^2.$$

Taking a union bound over $t$ and make use of the fact that $\sum_{t=1}^{\infty} t^{-2} < 2$ completes the proof. $\square$

We also have the following lemma:

**Lemma 36** (Analog of Lemma B.4 from Zhou et al. (2021)). *Under the same conditions as the previous lemma, with probability $1 - \frac{\delta}{2}$, we will have the following for all $\geq 1$ simultaneously:*

$$\sum_{s=1}^{t} \frac{\eta_s^2 w_s^2}{1 + w_s^2} \leq \frac{1}{4}\beta_t^2.$$

*Proof.* We still need to apply our Freedman's inequality (Corollary 33) to

$$\ell_s = \mathbb{E}\left[\frac{\eta_s^2 w_s^2}{1 + w_s^2}\bigg|\mathcal{F}_{s-1}\right] - \frac{\eta_s^2 w_s^2}{1 + w_s^2}.$$

As $\eta_s \mid \mathcal{F}_{s-1} \sim \mathrm{subG}(\sigma^2)$, $\eta_s^2 \mid \mathcal{F}_{s-1}$ is a sub-exponential random variable (Proposition 12) such that

$$\mathbb{E}\left[\exp(\lambda(\eta_s^2 - \mathbb{E}[\eta_s^2]))\big|\mathcal{F}_{s-1}\right] \leq \exp(16\lambda^2\sigma^4), \quad \forall |\lambda| \leq \frac{1}{4\sigma^2},$$

which consequently means

$$\mathbb{E}\left[\exp(\lambda\ell_s)|\mathcal{F}_{s-1}\right] \leq \exp\left(16\lambda^2\sigma^4(\min\{1, w_s^2\})^2\right), \quad \forall |\lambda| \leq \frac{1}{4\sigma^2}.$$

where the second step used the fact that $\frac{w_s^2}{1+w_s^2} \leq \min\{1, w_s^2\}$ as we used in the proof of Lemma 35. Again by Proposition 34, we can conclude that

$$\sum_{s=1}^{t} \sigma^4(\min\{1, w_s^2\})^2 \leq 2\sigma^2 d \ln\left(1 + \frac{tL^2}{d\lambda}\right).$$

Then, as we did in the proof of Theorem 13, we will apply Proposition 11 to the martingale difference sequence $\{\ell_s\}_{s=1}^{t}$ with $V_s = \sigma^4(\min\{1, w_s^2\})^2$, $f(\lambda) = 16\lambda^2$ for $\lambda < \frac{1}{4\sigma^2}$ and $f(\lambda) = \infty$ otherwise. Then for all $x, v > 0$ and $\lambda \in (0, \frac{1}{\sigma^2})$, we have

$$\Pr\left\{\sum_{s=1}^{t} \ell_s > x \wedge \sqrt{\sum_{i=1}^{n} \sigma_i^4(\min\{1, w_s^2\})^2} \leq v\right\} \leq \exp\left(-\lambda x + 16\lambda^2 v^2\right).$$

Picking $v^2 = \sum_{s=1}^{t} \sigma_i^4(\min\{1, w_s^2\})^2$ and $x = 4\sqrt{2}v\sqrt{\ln\frac{2}{\delta}}$ gives

$$\Pr\left\{\sum_{s=1}^{t} \ell_s > 4\sqrt{2\sum_{s=1}^{t} \sigma_i^4(\min\{1, w_s^2\})^2 \ln\frac{2}{\delta}}\right\} \leq \exp\left(-\frac{x^2}{32v^2}\right) = \frac{\delta}{2},$$

where $\lambda$ is set to $\frac{x}{32v^2} < \frac{1}{\sigma^2}$. Hence, with probability $1 - \frac{\delta}{4t^2}$, we indeed have

$$\sum_{s=1}^{t} \ell_s \leq 4\sqrt{2\sum_{s=1}^{t} \sigma_i^4(\min\{1, w_s^2\})^2 \ln\frac{2}{\delta}} \leq 8\sqrt{2\sigma^2 d \ln\left(1 + \frac{tL^2}{d\lambda}\right) \ln\frac{8t^2}{\delta}}.$$

Moreover, due to Proposition 34, we have

$$\sum_{s=1}^{t} \mathbb{E}\left[\frac{\eta_s^2 w_s^2}{1 + w_s^2}\bigg|\mathcal{F}_{s-1}\right] \leq \sigma^2 \sum_{s=1}^{t} \frac{w_s^2}{1 + w_s} \leq 2\sigma^2 d \ln\left(1 + \frac{tL^2}{d\lambda}\right),$$

which means, as $\frac{\eta_s^2 w_s^2}{1+w_s^2} \le \ell_s + \mathbb{E}[\frac{\eta_s^2 w_s^2}{1+w_s^2}]$,

$$\sum_{s=1}^{t} \frac{\eta_s^2 w_s^2}{1+w_s^2} \le 2\sigma^2 d\ln\left(1 + \frac{tL^2}{d\lambda}\right) + 8\sqrt{2\sigma^2 d\ln\left(1 + \frac{tL^2}{d\lambda}\right)\ln\frac{8t^2}{\delta}} \le \frac{1}{4}\beta_t^2.$$

Taking a union bound over all $t$ and again making use of the fact that $\sum_{t=1}^{\infty} t^{-2} < 2$ gives our conclusion. $\qquad\square$

Therefore, as long as their Lemmas B.3 and B.4 still hold, we can conclude exactly the same conclusion from their derivation. One may refer to their proof for the details. $\qquad\square$

## I  AUXILLIARY LEMMAS

**Lemma 37** (Self Bounding Inequality, Efroni et al. (2020, Lemma 38))**.** *Let $0 \le x \le a + b\sqrt{x}$ where $a, b, x \ge 0$, then we have*
$$x \le 4a + 2b^2.$$

*Proof.* As $x - b\sqrt{x} - a \le 0$, we have

$$\sqrt{x} \le \frac{b}{2} + \sqrt{\frac{1}{4}b^2 + 4a} \le \frac{b}{2} + \sqrt{\frac{b^2}{4}} + \sqrt{4a} = b + 2\sqrt{a}$$

from the fact that $\sqrt{a+b} \le \sqrt{a} + \sqrt{b}$. As $\sqrt{x} \ge 0$, we have

$$x \le (b + 2\sqrt{a})^2 \le 2b^2 + 4a$$

due to the relation that $(a + b)^2 \le 2a^2 + 2b^2$. $\qquad\square$

