# OpenReview forum: "Variance-Aware Sparse Linear Bandits"
_ICLR.cc/2023/Conference — ICLR 2023 poster_

### Official Review · Reviewer_br26 · 2022-10-21

**Confidence:** 4
**Clarity, Quality, Novelty And Reproducibility:** good.
**Correctness:** 3
**Technical Novelty And Significance:** 2
**Empirical Novelty And Significance:** Not applicable
**Recommendation:** 5

**Strength And Weaknesses:**

It is interesting to study sparse linear bandits from variance-aware prospective. However, I feel the regime that variance-aware algorithm can improve for sparse linear bandits is not sufficiently interesting. For the current regret bound, it will be dimension-free only for the deterministic case (sigma=0) or nearly deterministic case (sigma=1/sqrt{d}). Sparse linear models or bandits are motivated by the use of high-dimensional features so d is typically very large. The requirement for sigma=1/sqrt{d} is very restricted and does not match any practical application. Thus, the novelty beyond variance-aware linear bandits is limited.

Second, I feel a variance-aware or problem-dependent lower bound is missing. I would like to see how this problem depends on noise-variance fundamentally. Existing minimax lower bound cannot explain this.


**Summary Of The Paper:**

This paper studied sparse linear bandits with noise variance appearing in the regret bound.

**Summary Of The Review:**

Although a new prospective for sparse linear bandits is proposed, the improved regime is less interesting. A lower bound to fully characterize the problem is missing.

---

> ### Author Response · Authors · 2022-11-16
> **Response to Reviewer br26**
>
> Thank you for your insightful comments! Here are our responses:
>
> ---
>
> **Significance of Our Result:**
>
> Thank you for discussing this. Indeed, our deterministic-case regret is only independent of $d$ and $T$ when the environment is nearly-deterministic ($\sigma_t\lesssim \frac{1}{\sqrt d}$ for most of $t$'s). However, we would like to emphasize that **our significant improvement not only lies in the nearly-deterministic case but in fact, all sparse cases where $s\ll d$**.
>
> First, emphasizing our algorithm's superiority in the nearly-deterministic case is mainly due to our motivating example. Our result significantly generalizes the divide-and-conquer algorithm for deterministic cases by allowing gentle transition from the $\widetilde{\mathcal O}(\sqrt{dT})$ worst-case regret to the $\widetilde{\mathcal O}(1)$ best-case regret. We note that the result of this regime is only one part of our contributions.
>
> To illustrate the significance of our algorithm, let us consider a general **sparse linear bandit in a heteroscedastic environment ($\sigma_t$ is different for different $t$)**. Without our algorithm, there are two ways of handling it: either directly applying a variance-aware linear bandit algorithm by Zhou et al. (2021), Zhang et al. (2021), or Kim et al. (2021), or using a non-variance-aware (i.e., minimax) sparse linear bandit algorithm by Carpentier and Munos (2012) or Abbasi-Yadkori et al. (2012). However, **neither of them can be as good as ours**:
> 1. Compared with directly applying a variance-aware linear bandit algorithm, our bound reduces the dependency on $d$ by at least $\sqrt d$, i.e., from $d\sqrt T$ (Zhou et al., 2021) or $d^{1.5}\sqrt T$ (Kim et al., 2021) to $\sqrt{dT}$. In sparse linear bandit problems, $d$ is often extremely large while $s\ll d$ is small (also mentioned by Reviewer MSEd). Hence, **this improvement of $\sqrt d$ is significant**.
> 2. Compared with a non-variance-aware sparse linear bandit algorithm, our bound is also dominating due to its **instance-dependent nature**. For example, suppose there is only small noise except for no more than $T'\ll T$ rounds, where the noises $\sigma_t\approx 1$. This can happen due to a sudden burst of noises in the measurements. In such cases, our bound is roughly $\sqrt{dT'}$ (again ignoring dependencies on $s\ll d$), which is significantly better than the $\sqrt{dT}$ bound directly yielded by non-variance-aware algorithms.
>
> Thus, our bound has an adequate contribution compared with previous ones.
>
> ---
>
> **Variance-aware Lower Bound:**
>
> Thank you for mentioning the lower bound. A variance-aware or instance-dependent lower bound is fascinating but also highly non-trivial --- indeed, in variance-aware linear bandits (which is a particular case of ours when setting $s=d$), such a lower bound is also absent. Here, we briefly sketch why we cannot direct adapt existing minimax lower bounds to variance-aware ones.
>
> Suppose we follow the proof of a typical minimax lower bound for sparse linear bandits (let's assume $s=1$ for simplicity). We will consider $d$ possible instances $\theta^{(0)},\theta^{(2)},\ldots,\theta^{(d)}$ such that $\theta^{(i)}=\epsilon \vec e_i$ where $\epsilon$ is some parameter to be tuned. Then we consider the KL divergences between $\theta^\ast=\theta^{(i)}$ and $\theta^\ast=\vec 0$, giving (see, e.g., Theorem 3.1 of Antos and Szepesvári (2009))
>
> $$
> \begin{equation*}
> \left (T-\frac Td-cT\epsilon \sqrt{\frac 1d\sum_{t=1}^T \sigma_t^{-2}}\right )\epsilon,\tag{a}
> \end{equation*}
> $$
>
> where $c\ll 1$ is a constant. For an $\Omega\left (\sqrt{d\sum_{t=1}^T \sigma_t^2}\right )$-style lower bound, we must set $\epsilon=\sqrt dT^{-1}\sqrt{\sum_{t=1}^T \sigma_t^2}$, making the last term in Eq. (a)
>
> $$
> cT\epsilon^2 \sqrt{\frac 1d\sum_{t=1}^T \sigma_t^{-2}}=\frac{\sqrt d}{T}\left (\sum_{t=1}^T \sigma_t^2 \right )\sqrt{\sum_{t=1}^T \sigma_t^{-2}}.
> $$
>
> Although this term can be well-bounded by $\sigma \sqrt{dT}$ when $\sigma_i\equiv \sigma$, it can explode to $\infty$ in the simple case that $\sigma_1=1$, $\sigma_2=0$. (Note that the Cauchy-Schwartz gives $\sqrt{(\sum_{t=1}^T \sigma_t^2)(\sum_{t=1}^T \sigma_t^{-2})}\ge T$ instead of $\le T$.) As a single $0$ can make the whole analysis break, it is non-trivial to give a lower bound in heteroscedastic cases; we have listed variance-aware lower bounds as part of future work. Many thanks for your discussion!
>
> ---
>
> We thank you once again for such a valuable review of our paper. We are more than happy to answer any further questions.

---

### Official Review · Reviewer_babC · 2022-10-30

**Confidence:** 4
**Correctness:** 4
**Technical Novelty And Significance:** 3
**Empirical Novelty And Significance:** Not applicable
**Recommendation:** 8

**Clarity, Quality, Novelty And Reproducibility:**

The paper is well written with the key ideas and proof sketches shown in the main body of the paper . It proposes a novel black-box framework for plugging in linear bandit algorithms and obtain sparse ones. Also, it is self-contained with all the relevant results.

**Strength And Weaknesses:**


   Pros:

   (A) The generic formulation uses gap-dependent rounds split into commit and explore rounds to delicately balance exploration while minimizing the resulting regret. The resulting regret terms are almost tight except for a square root sparsity (s) factor which is typically small in practice.
   (B) Quite a few state-of-the-art algorithms such as VOFUL2 and Weighted OFUL are considered in the framework and their regret are rigorously derived for both the
framework and the corresponding algorithm components.

Cons:

   (i) Experiments show-casing the proposed regret bounds on datasets would have made the paper much more compelling.
   (ii) As noted in the paper, there is still a gap with respect to the lower bounds and the obtained bounds.



**Summary Of The Paper:**

The paper provides a black-box conversion from variance-aware linear bandit algorithm to its corresponding variance-aware sparse linear bandit algorithm. The regret framework is applied to two recent algorithms and obtain quite compelling regret bounds.

**Summary Of The Review:**

Overall, a good theoretical paper for providing new algorithms for the variance-aware sparse linear bandits and would be of great interest to the community.

---

> ### Author Response · Authors · 2022-11-16
> **Response to Reviewer babC**
>
> Thank you for your insightful comments! Here are our responses:
>
> ---
>
> **The Dependency on $s$:**
>
> Thanks for mentioning this! Unfortunately, our current algorithm cannot give a refined bound without much modification. Consider the special case where $T=ds^2$, $\theta_i^\ast=\frac{1}{\sqrt s}$ for all $i$ (recall that $\lVert \theta^\ast\rVert_2\le 1$) and $\sigma_t\equiv 1$ for all $t$. Then for each coordinate $i$, the total suffered regret due to it is $\sum_{i=1}^s \sum_{\Delta\ge \theta_i^\ast} n_\Delta^b (\theta_i^\ast)^2$. In our algorithm, we have (omitting all constants and dependencies on $\log \frac 1\delta$)
>
> $$
> n_\Delta^b \approx \frac{1}{\Delta}\sqrt{n_\Delta^b\left (1+\frac{d}{\Delta^2}\right )},
> $$
> where we plugged the fact that $(r_{k,i}-\mathbb{E}[r_{k,i}])^2\le (1+\frac{4d}{\Delta^2}\sigma_k^2)$ (top of Page 8) into Eq. (2). Hence, $n_\Delta^b\lesssim \frac{d}{\Delta^4}$, giving
> $$
> \sum_{i=1}^s \sum_{\Delta\ge \theta_i^\ast} n_\Delta^b (\theta_i^\ast)^2\lesssim d\sum_{i=1}^s (\theta_i^\ast)^{-2}=ds^2=s\sqrt{dT}.
> $$
>
> Therefore, in the worst case, our algorithm has to suffer $s\sqrt{dT}$ regret.
>
> On the other hand, we would like to mention that the current paper mainly focuses on the dependencies on $d$ and $\sum_{t=1}^T \sigma_t^2$ --- in sparse linear bandit problems, $d$ and $T$ are often extremely large (also mentioned by Reviewers MSEd and br26). Indeed, our framework successfully ensures tight (up to logarithmic factors) dependencies on both $d$ and $T$, namely $\widetilde{\mathcal O}(\sqrt{dT})$ in the worst case and $\widetilde{\mathcal O}(1)$ in the deterministic case. Improving the dependency on $s$ is left as an important future work.
>
>
> ---
>
> **Experiments:**
>
> Thank you for your suggestion! Recall that we propose a general reduction framework whose evaluation requires an efficient implementation of the plug-in linear bandit algorithm. Sadly, we can implement neither the `Weighted-OFUL` algorithm by Zhou et al. (2021), which requires calculating a minimization problem in the confidence ball nor the `VOFUL2` algorithm by Kim et al. (2021), which is computationally intractable.
>
> Generally speaking, confidence-set-based linear bandit algorithms all require an optimization oracle over the confidence set (see, e.g., Abbasi-Yadkori et al. (2011, 2012); Zhou et al. (2021)); they did not provide experiments with large $d$ as well). Unfortunately, this oracle cannot be implemented efficiently (note that Abbasi-Yadkori et al. (2011) only provided experiments when $d=2$, where grid search is possible; on the contrary, in sparse linear bandits where $s\ll d$, we can never use a tiny $d$ like they did).
>
> ---
>
> We thank you once again for such a valuable review of our paper. We are more than happy to answer any further questions.

---

### Official Review · Reviewer_MSEd · 2022-11-04

**Confidence:** 3
**Correctness:** 4
**Technical Novelty And Significance:** 3
**Empirical Novelty And Significance:** Not applicable
**Recommendation:** 6

**Clarity, Quality, Novelty And Reproducibility:**

The motivation and technical contribution is clearly stated, and most part of the paper is easy to follow.
The theoretical results look intutive and sound, though I didn't carefully check the proof.

**Strength And Weaknesses:**

Strength

The idea of extending variance-aware algorithm to sparse linear bandit is well-motivated.
The proposed techniques to make explore-then-commit framework to work when the desired regret is unknown (as the regret for variance-aware algorithms depends on the variances, which are unknown) may are intuitive and technically interesting.

Weakness

1. The description in Section 3.1 can be further improved. Its current form is a bit hard to follow, e.g., maybe first define what is arm $i$ in the first paragraph on page 6.

2. I would appreciate it if the authors can provide more explanations about the fundamental reasons for requiring action set to be unit sphere for all time steps, while Abbassi et. al. (2012) does not? It seems footnote 3 only provides an extreme case, where $\Omega(d)$ is unavoidable, i.e., unit sphere is only a sufficient condition to help avoid this. Is it possible to further relax this action set assumption, and allow for drifting action set?

3. The authors mentioned the data-poor regimes where $d>>T$ is beyond the scope of this paper. Can the authors elaborate on what are the possible difficulties in this case, and the reason why the proposed framework does not apply?

4. The authors mentioned the $O(\sqrt{s})$ gap to the regret lower bound can be improved if better variance aware linear bandit algorithm exists. Can the author provide more justification for this, e.g. why this is due to the sub-optimality of the adopted variance aware linear bandit algorithm, instead of the proposed explore-then-commit framework. And is it possible to derive a regret for the proposed framework (in variance aware sparse bandit), that depends on the regret of the adopted variance aware linear bandit, so that, it directly shows with an optimal variance aware linear bandit, the proposed framework is able to close the $O(\sqrt{s})$ gap?


**Summary Of The Paper:**

This paper proposes the first variance aware algorithms for sparse linear bandits, which interpolates between the $\sqrt{dT}$ bound in worst case when variances are constant, and the bound independent of $d$ in the deterministic case where variances are 0.
The authors achieved this by an explore-then-commit framework that can be applied to existing variance-aware linear bandit algorithms, and showed that by plugging in VOFUL, it can match the regret lower bound for the worse case by a factor of $\sqrt{s}$, and attain regret upper bound $O(s^3)$ in the deterministic case, i.e., removed the dependence on d and T.

**Summary Of The Review:**

Variance aware algorithm for sparse linear bandit algorithm is well-motivated, and this paper provides the first solution via a explore-then-commit framework that converts existing variance aware algorithm to the sparse setting. However, there is still an $O(\sqrt{s})$ gap to the regret lower bound in the worst case, and it is not very clear to me whether this gap can be closed under the current framework.

---

> ### Author Response · Authors · 2022-11-16
> **Response to Reviewer MSEd**
>
> Thank you for your insightful comments! Here are our responses:
>
> ---
>
> **Assumption on the Action Set:**
>
> Thanks for pointing out this. Such an assumption is due to our usage of random projection. More preciously, as the random projection procedure may play $(\pm a,\pm a,\ldots,\pm a)$ for any possible $a$, we must ensure the action set is spherically symmetric. Thus, the most natural choice of the action set is just the unit sphere $\mathbb{S}^{d-1}$ -- this is also the choice of Carpentier and Munos (2012).
>
> For an arbitrary action set, we suspect an OFUL-style algorithm (Abbasi-Yadkori et al., 2012) is needed instead of our explore-then-commit-style algorithm (which is easy to analyze).
>
> ---
>
> **Why not Considering Data-poor Regimes:**
>
> This is a good question! Indeed, as illustrated by Hao et al. (2020), data-poor regimes are fundamentally different from data-rich ones in the sense that the (minimax) lower bound is of order $\Omega(d^{1/3}s^{1/3}T^{2/3})$ (see their Theorem 3.3, Definition 3.1, and Remark 3.2) instead of $\Omega(\sqrt{sdT})$. Hence, our current algorithm, whose regret is of order $\mathcal O(s\sqrt{dT})$, can never remain valid in data-poor regimes.
>
> The most critical difference is that exploration is much harder in data-poor regimes. For example, we even cannot finish the very first random projection step of our algorithm (i.e., "explore" phase), because it will take $n_1^b\approx d\gg T$ rounds when $\sigma_t\equiv 1$ as $(r_{t,i}-\mathbb{E}[r_{t,i}])^2\approx (1+\frac{d}{\Delta^2}\sigma_t^2)$ (top of Page 8). Hence, algorithms in data-poor regimes must follow a completely different algorithm design mechanism. That is why we say it is beyond the scope of our paper.
>
> ---
>
> **Closing the Gap with the Lower Bound:**
>
> Thank you for mentioning this. Our claim about better regret is in the deterministic case instead of the worst case. Informally, a variance-aware linear bandit algorithm which ensures
>
> $$
> \widetilde{\mathcal O}\left (f(d)\sqrt{\sum_{t=1}^T \sigma_t^2}+g(d)\right )
> $$
>
> regret (ignoring dependencies on $\log \frac 1\delta$) can result in an algorithm for sparse linear bandits with regret guarantee
>
> $$
> \widetilde{\mathcal O}\left ((s\times f(s)+s\sqrt d)\sqrt{\sum_{t=1}^T \sigma_t^2}+(s\times g(s)+s)\right )
> $$
>
> by using our proposed framework (as remarked after Definition 1). Hence, the bottleneck of the worst-case regret, namely $s\sqrt{dT}$, is due to our framework (Theorem 3). Instead, only the bottleneck of the best-case regret, namely $s\times g(s)$, is due to the plug-in linear bandit algorithm.
>
> We agree that our description at the end of "Our Contributions" can be misleading; we only meant that we could improve the deterministic-case regret by deploying a better variance-aware linear bandit algorithm. We have enhanced it in the revision (highlighted in blue).
>
> At last, we would like to mention that the current paper only focuses on the dependencies on $d$ and $T$ (i.e., the sum of variances) as in sparse linear bandit problems, $d$ and $T$ are often extremely large (also mentioned by Reviewer br26) while $s\ll d$ is negligible in practice. Indeed, our framework successfully ensures tight (up to logs) dependencies on both $d$ and $T$, namely $\widetilde{\mathcal O}(\sqrt{dT})$ in the worst case and $\widetilde{\mathcal O}(1)$ in the deterministic case. Improving the dependency on $s$ is left as an important future work.
>
> ---
>
> **Writing:**
>
> Thank you for mentioning this. The "arm $i$" in the first paragraph of Page 6 means an arbitrary coordinate that previous "explore" phases have identified. We have updated our manuscript according to your suggestion (highlighted in blue).
>
> ---
>
> We thank you once again for such a valuable review of our paper. We are more than happy to answer any further questions.

---

### Official Review · Reviewer_DQpC · 2022-11-05

**Confidence:** 3
**Correctness:** 4
**Technical Novelty And Significance:** 4
**Empirical Novelty And Significance:** Not applicable
**Recommendation:** 8

**Clarity, Quality, Novelty And Reproducibility:**

Clarity - overall the writing was clear

Novelty - while there has been work on variance-aware linear bandits and separately work on sparse linear bandits, this work appears novel in studying variance-aware methods for sparse linear bandits.

Reproducibility -- the authors include proof sketches in the main paper and appendices with technical details



**Strength And Weaknesses:**

Strengths
- The authors show that variance-aware methods for linear bandits can be adapted variance-aware methods in the sparse linear bandit setting.
- While the authors employ standard explore-then-commit style approach in identifying large coefficients for sparse linear bandits, for the variance aware setting the authors use loop over explore-then-commit steps adaptively changing the threshold.
- The presentation and writing were mostly clear and easy to follow.

Weaknesses
- (minor) Some experiments (even for a few toy samples) could help clarify whether empirically (a) the variance-aware linear bandit methods perform poorly in sparse settings compared to the proposed framework’s adaptation of those methods, (b) whether the dependence on $s$ in the bound is tight (i.e. would a new algorithm be needed or possibly just an alternative proof), and (c) if the empirical regret incurred by the proposed framework when actually applied to (nearly) deterministic settings is close to what could be achieved by a divide-and-conquer method designed specifically for the deterministic setting.


**Summary Of The Paper:**

The authors propose a variance-aware framework for stochastic sparse linear bandit problems.  The performance (as captured by regret bounds) bridges the general stochastic reward setting and the deterministic reward setting, which can be efficiently solved by a divide-and-conquer method.  The proposed framework employs variance-aware linear bandit methods (i.e. methods that did not assume or exploit sparsity of the coefficient vector), using repeated ‘explore’ steps and ‘commit’ steps with a adapting threshold.  The authors prove regret bounds using two example variance-aware linear bandit methods as sub-routines and achieve ambient dimension free bounds for the deterministic case.

**Summary Of The Review:**

The problem is novel (though related to studied problems), the approach extends strategies (explore-then-commit for sparse linear bandits; variance-aware linear bandit methods) in a non-trivial manner.  The paper is well-written.

---

> ### Author Response · Authors · 2022-11-16
> **Response to Reviewer DQpC**
>
> Thank you for your insightful comments! Here are our responses:
>
> ---
>
> **The Dependency on $s$:**
>
> Thanks for mentioning this! Unfortunately, our current algorithm cannot give a refined bound without much modification. Consider the special case where $T=ds^2$, $\theta_i^\ast=\frac{1}{\sqrt s}$ for all $i$ (recall that $\lVert \theta^\ast\rVert_2\le 1$) and $\sigma_t\equiv 1$ for all $t$. Then for each coordinate $i$, the total suffered regret due to it is $\sum_{i=1}^s \sum_{\Delta\ge \theta_i^\ast} n_\Delta^b (\theta_i^\ast)^2$. In our algorithm, we have (omitting all constants and dependencies on $\log \frac 1\delta$)
>
> $$
> n_\Delta^b \approx \frac{1}{\Delta}\sqrt{n_\Delta^b\left (1+\frac{d}{\Delta^2}\right )},
> $$
> where we plugged the fact that $(r_{k,i}-\mathbb{E}[r_{k,i}])^2\le (1+\frac{4d}{\Delta^2}\sigma_k^2)$ (top of Page 8) into Eq. (2). Hence, $n_\Delta^b\lesssim \frac{d}{\Delta^4}$, giving
> $$
> \sum_{i=1}^s \sum_{\Delta\ge \theta_i^\ast} n_\Delta^b (\theta_i^\ast)^2\lesssim d\sum_{i=1}^s (\theta_i^\ast)^{-2}=ds^2=s\sqrt{dT}.
> $$
>
> Therefore, in the worst case, our algorithm has to suffer $s\sqrt{dT}$ regret.
>
> On the other hand, we would like to mention that the current paper mainly focuses on the dependencies on $d$ and $\sum_{t=1}^T \sigma_t^2$ --- in sparse linear bandit problems, $d$ and $T$ are often extremely large (also mentioned by Reviewers MSEd and br26). Indeed, our framework successfully ensures tight (up to logarithmic factors) dependencies on both $d$ and $T$, namely $\widetilde{\mathcal O}(\sqrt{dT})$ in the worst case and $\widetilde{\mathcal O}(1)$ in the deterministic case. Improving the dependency on $s$ is left as an important future work.
>
>
> ---
>
> **Experiments:**
>
> Thank you for your suggestion! Recall that we propose a general reduction framework whose evaluation requires an efficient implementation of the plug-in linear bandit algorithm. Sadly, we can implement neither the `Weighted-OFUL` algorithm by Zhou et al. (2021), which requires calculating a minimization problem in the confidence ball nor the `VOFUL2` algorithm by Kim et al. (2021), which is computationally intractable.
>
> Generally speaking, confidence-set-based linear bandit algorithms all require an optimization oracle over the confidence set (see, e.g., Abbasi-Yadkori et al. (2011, 2012); Zhou et al. (2021)); they did not provide experiments with large $d$ as well). Unfortunately, this oracle cannot be implemented efficiently (note that Abbasi-Yadkori et al. (2011) only provided experiments when $d=2$, where grid search is possible; on the contrary, in sparse linear bandits where $s\ll d$, we can never use a tiny $d$ like they did).
>
> ---
>
> We thank you once again for such a valuable review of our paper. We are more than happy to answer any further questions.

---

### Author Response · Authors · 2022-11-16
**Thank You Note to All the Reviewers**

Dear reviewers,

We thank you a lot for your valuable feedback, which we benefit greatly from. Many thanks for noting that "the presentation and writing were mostly clear and easy to follow" (Reviewers DQpC, MSEd, and babC); our problem setting is "novel," "well-motivated," and is "a new perspective" (Reviewers DQpC, MSEd, and br26); our approach "extends existing ones in a non-trivial manner" and "may be intuitive and technically interesting" (Reviewers DQpC and MSEd); our near-optimal result is "of great interest to the community" (Reviewer babC).

We have revised our manuscript according to your suggestions. Thank you all for your effort!

Best regards,
Authors

---

### Decision · Program_Chairs · 2023-01-20

**Decision:**

Accept: poster

**Justification For Why Not Higher Score:**

While the reviewers mostly agree with the novelty of proposed method and its theoretical contribution, there is one remaining concern shared by reviewers regarding the restricted and impractical assumption of unit sphere action set.

**Justification For Why Not Lower Score:**

All reviewers agree that studying sparse linear bandits from the variance-aware perspective is novel and interesting. The improved regret is a good theoretical contribution.

**Metareview: Summary, Strengths And Weaknesses:**

The authors propose a variance-aware framework for sparse linear bandits. The framework follows explore-then-commit method and employs existing variance-aware linear bandit algorithms using repeated ‘explore’ steps and ‘commit’ steps with an adapting threshold. The authors proved tighter regret bounds of the proposed framework with two example variance-aware linear bandit methods, VOFUL2 and Weighted OFUL. Specifically, the regret bounds are dimension-free when the environment is nearly deterministic. All reviewers appreciate the novel perspective of variance-aware sparse linear bandits. One shared concern by the reviewers is the unit sphere assumption of action set: this assumption is acceptable from a theoretical perspective but can be very restrictive in practice. The authors are suggested to further discuss this assumption in the final version. The authors are also encouraged to include additional discussion on $O(\sqrt{s})$ gap between regret lower and upper bound and the problem-dependent lower bound. Overall, I am happy to recommend acceptance.



**Note From Pc:**

if the above contains the word "oral" or "spotlight" please see: "oral" presentation means -> notable-top-5% and "spotlight" means -> notable-top-25%. As stated in our emails, we are disassociating presentation type from AC recommendations